# Effects of fertilization and stand age on $N_2O$ and NO emissions from tea plantations: A site-scale study in a subtropical region using a modified biogeochemical model

Wei Zhang [1], Zhisheng Yao [1], Xunhua Zheng [1, 2], Chunyan Liu [1], Rui Wang [1], Kai Wang [1], Siqi Li [1], Shenghui Han [1], Qiang Zuo [3], Jianchu Shi [3]

[1] State Key Laboratory of Atmospheric Boundary Layer Physics and Atmospheric Chemistry, Institute of Atmospheric Physics, Chinese Academy of Sciences, Beijing 100029, P. R. China

[2] University of Chinese Academy of Sciences, Beijing 100049, P.R. China

[3] College of Resources and Environmental Sciences, China Agricultural University, Beijing 100193, P.R. China

*Corresponding to*: Xunhua Zheng (xunhua.zheng@post.iap.ac.cn); Zhisheng Yao (zhishengyao@mail.iap.ac.cn)

**Abstract.** To meet increasing demands, tea plantations are rapidly expanding in China. Although the emissions of nitrous oxide ($N_2O$) and nitric oxide (NO) from tea plantations may be substantially influenced by soil pH reduction and intensive nitrogen fertilization, process model-based studies on this issue are still rare. In this study, the process-oriented biogeochemical model, Catchment Nutrient Management Model - DeNitrification-DeComposition (CNMM-DNDC), was modified by adding tea growth-related processes that may induce a soil pH reduction. Using a dataset for intensively managed tea plantations at a subtropical site, the performances of the original and modified models for simulating the emissions of both gases subject to different fertilization alternatives and stand ages were evaluated. Compared with the observations in early stage of a tea plantation, the original and modified models showed comparable performances for simulating the daily gas fluxes (with Nash-Sutcliffe index (NSI) of 0.10 versus 0.18 for $N_2O$ and 0.32 versus 0.33 for NO), annual emissions (with NSI of 0.81 versus 0.94 for $N_2O$ and 0.92 versus 0.94 for NO) and annual direct emission factors ($EF_d$S). For the modified model, the observations and simulations demonstrated that short-term replacement of urea with oilcake stimulated $N_2O$ emissions by ~62% and ~36% and mitigated NO emissions by ~25% and ~14%, respectively. The model simulations resulted in a positive dependence of $EF_d$ of either gas against nitrogen doses, implicating the importance of model-based quantification of this key parameter for inventory. In addition, the modified model with pH-related scientific processes showed overall inhibitory effects on the gases emissions in the mid to late stages during a full tea lifetime. In conclusion, the modified CNMM-DNDC exhibits the potential for quantifying $N_2O$ and NO emissions from tea plantations under various conditions. Nevertheless, wider validation is still required for simulation of long-term soil pH variations and emissions of both gases from tea plantations.

# 1 Introduction

Tea (*Camellia sinensis* (L.) Kuntze), as a perennial cash crop, has been widely cultivated long-term in the tropical and subtropical regions of the world, with nearly 90% of the global tea harvest area currently located in Asia and over 50% of that located within China (http://www.fao.org/faostat/). To maximize the economic benefits, especially in China, tea production has expanded intensively, mostly through the conversion of arable uplands, rice paddies and forests into tea plantations (e.g., Xue et al., 2013; Yao et al., 2015). For instance, both the total harvest area and production have dramatically increased by 166% (from $1.09 \times 10^6$ to $2.90 \times 10^6$ ha) and 253% (from $6.8 \times 10^5$ to $2.40 \times 10^6$ Mg), respectively, from 2000 to 2016 (http://www.fao.org/faostat/).

As a leaf/bud-harvested crop, nitrogen is the key nutrient for yield. Thus, high tea yields are largely supported by the intensive application of nitrogen fertilizers. The nitrogen inputs amount to 450–1200 (mean: 553) kg N ha$^{-1}$ yr$^{-1}$ in the primary areas of tea cultivation in China (Han et al., 2013), which is much higher than the recommended doses of 250–375 kg N ha$^{-1}$ yr$^{-1}$ (Fu et al., 2012; Hirono and Nonaka, 2012, 2014; Hou et al., 2015; Li et al., 2016; Tokuda and Hayatsu, 2004; Yamamoto et al., 2014; Yao et al., 2015, 2018). This intensive nitrogen application results in superfluous reactive nitrogen remaining in the soil. The excessive reactive nitrogen induces the high potential for nitrous oxide ($N_2O$) and nitric oxide (NO) emissions, thus leading to the detrimental consequences of global warming and air pollution.

The tea plant has been well known as one of the very few families tolerable to high levels of aluminum ion ($Al^{3+}$) and thus can grow well in acidic soil (Taylor, 1991). Mature leaves of the tea plant may contain up to 30 g Al kg$^{-1}$ on dry weight basis (Matsumoto et al., 1976) without experiencing Al toxicity (Morita et al., 2008). Part of the tissue Al further returns to soil through plant trimming and thus leads to Al accumulation in surface soil of a tea plantation. In addition, the Al under an acidic condition can be recombined with the organic matter derived from root exudation. This process further facilitates the accumulation of Al in the supper soil layer of a tea plantation (Lin et al., 2014). The former mechanism of Al accumulation in surface soil almost does not occur for the absolutely majority of plant families that much more weakly absorption of Al than tea plant (Taylor, 1991; Matsumoto et al., 1976). Hence, the soil pH of tea plantations decreases with the increased stand age jointly due to the processes of (i) acid release by root exudation and (ii) hydrogen ion (H$^+$) production in the hydrolysis of the accumulated $Al^{3+}$ from residue decomposition in surface soil. The high nitrogen doses combined with the decreased soil pH may further promote the production of the harmful nitrogenous gases through both microbial processes (e.g., nitrification and denitrification) and non-biological mechanisms (e.g., chemo-denitrification) (e.g., Chen et al., 2017; Fu et al., 2012; Yao et al., 2018), especially for the low pH. A number of field studies have demonstrated that much more $N_2O$ and NO were emitted from tea plantations than those in other upland fields (e.g., Akiyama et al., 2006). In China, the $N_2O$ and NO emissions from tea plantations are 16.6 and 14.9 kg N ha$^{-1}$ yr$^{-1}$ on average, respectively (Fu, 2013; Han et al., 2013; Yao et al., 2015, 2018). In 2013, for instance, the $N_2O$ emissions from tea plantations in China accounted for more than one-tenth of the national total emissions of this gas from croplands and contributed to 85% of the total $N_2O$ emissions from global tea plantations (Li et al., 2016). To alleviate the negative impacts on environmental quality and human health,

organic fertilization has been strongly recommended in China and adopted in nearly $4.5 \times 10^4$ ha of tea fields by 2011 (Han et
al., 2013). Application of organic fertilizers in tea fields can improve soil fertility (e.g., Han et al., 2013), while stimulating
$N_2O$ emissions but mitigating NO release (Yao et al., 2015). Therefore, investigating the impacts of replacing synthetic
nitrogen fertilizer with organic manure and the effect of stand ages on the emissions of $N_2O$ and NO from tea plantations is
necessary for understanding the mechanisms of nitrogen cycling and effectively mitigating the emissions of both of the
nitrogenous gases from tea fields.
Compared with time- and labor-consuming field experiments, from which first-hand information of $N_2O$ and NO
emissions could be obtained, modeling approaches based on sufficient validation have been proposed to overcome the limits
of field measurements (e.g., Chen et al., 2017). Because process-oriented biogeochemical models such as DNDC (e.g., Li,
2000), LandscapeDNDC (e.g., Haas et al., 2012), WNMM (e.g., Li et al., 2007) and CNMM-DNDC (Zhang et al., 2018) are
generally designed following the basic theories of physics, chemistry, physiology and biology, they are expected to be
widely applicable under various climates, soils, land uses and field management practices. These models, in principle, can
facilitate the understanding of the interactions among various processes, identify gaps in current knowledge, and
temporally/spatially extrapolate the results from experiments (Chen et al., 2008). Among these models, the Catchment
Nutrient Management Model - DeNitrification-DeComposition (CNMM-DNDC) is one of the latest versions of the DNDC.
CNMM-DNDC was established by incorporating the core carbon and nitrogen biogeochemical processes of DNDC
(including the processes of decomposition, nitrification, denitrification and fermentation) into the hydrologic framework of
the CNMM, and it therefore inherited the features from both CNMM and DNDC (Zhang et al., 2018). CNMM-DNDC was
established to solve a common bottleneck problem of most biogeochemical models, i.e., the inability to simulate the lateral
flows of water and nutrients. This solution potentially enables the model to identify the best management practices of
intensive cropping systems. In its initial validation in a catchment with calcareous soils and complex landscapes, the
CNMM-DNDC performed fairly well in simulating ecosystem productivity (represented by crop yields in croplands),
hydrological nitrogen losses by soil leaching and nitrate discharge in streams, and emissions of gaseous carbon (carbon
dioxide, methane) and nitrogenous gases ($N_2O$, NO and ammonia) from different lands (forests and arable lands cultivated
with maize, wheat, oil rape, or paddy rice) (Zhang et al., 2018). However, the scientific processes of soil pH reduction due to
tea growth is still lacking in the CNMM-DNDC. This gap may induce significant biases in simulating the fluxes of both
nitrogenous gases from tea fields, especially for long term prediction, because soil pH is the key factor regulating $N_2O$ and
NO emissions from the soil (e.g., Chen et al., 2017; Yao et al., 2018). Therefore, the authors hypothesize that adding the
missing scientific processes which lead to soil pH reduction into the internal model program codes can improve the
performance of the CNMM-DNDC in simulating the $N_2O$ and NO emissions from tea plantations with different stand ages.
Filling the gap in the model is especially necessary for predicting the long term emissions of both gases from tea plantations.
To test the above hypothesis, the authors conducted a case study using a unique experimental dataset, which was
obtained by Yao et al. (2015, 2018) in a tea plantation with field treatments of fertilization alternatives and stand ages. The
aims of this case study were to (i) attempt to fill the gap in the CNMM-DNDC through addition of the processes that may
induce soil pH reduction due to tea growth, (ii) compare the performances of original and modified models in simulating
$N_2O$ and NO emissions, and (iii) evaluate the modified model performance in simulating the direct emission factors ($EF_d$s) of
different annual nitrogen doses, and the $N_2O$ and NO emissions affected by short-term replacement of a widely applied
synthetic nitrogen fertilizer (urea) with a typical organic manure (oilcake) and by the stand ages within the early stage (1−6
years) of a new tea plantation.

## 2 Materials and methods

### 2.1 Introduction to the field site and experimental treatments

The field site (32°7.37′N, 110°43.18′E, 441 m above sea level) selected for this modeling case study is located in
Fangxian county, Hubei province, China. The field site is subject to a northern subtropical monsoon climate, with annual
precipitation of 914 mm and a mean air temperature of 14.2 °C in 2003−2011 (Yao et al., 2015). Two plots at the field site
were involved in this study, encoded as T08 and T12, respectively. Both lands had been consecutively long-term cultivated
with paddy rice in summer and upland crops (or drained but fallowed) in winter until tea seedlings were transplanted in
March 2008 for T08 or March 2012 for T12. Conventional fertilization practices had been adopted in both plots. A typical
synthetic fertilizer (urea) was regularly applied at 450 (150 in autumn and 300 in spring) kg N ha$^{-1}$ yr$^{-1}$ (encoded as T08-UN
and T12-UN). To determine the annual $EF_d$ (the fraction of the applied fertilizer nitrogen released in the form of $N_2O$ or NO
within the one-year period after fertilization) of either gas and to investigate the effects of short-term synthetic fertilizer
replacement by organic manure on $N_2O$ and NO emissions, eight spatially replicated subplots were randomly set in either
T08 or T12: four for the control without nitrogen fertilizer applied (NN) and the others for exclusive application of organic
manure (OM) in 2013 (only T08) and 2014 (both T08 and T12). Each daily flux was inferred from the single measurement
based on five gas samples from a 30-min enclosure of a static opaque chamber between 09:00 and 11:00 (Beijing time).
Oilcake, one of the most widely applied organic manures in the subtropical regions of China, was exclusively amended in
the OM subplots to fully replace the urea, and nitrogen doses with the urea application outside the NN and OM subplots of
either plot. The NN and OM treatments were encoded as T08-NN, T08-OM, T12-NN, and T12-OM. T08-NN and T08-OM
were adopted consecutively in two full years (from October 2012 to March 2014), and T12-NN and T12-OM in one full year
(from October 2013 to March 2014). The organic manure in dry weight contained 7.1% nitrogen and 43.3% carbon. The
topsoil (0−15 cm depth) of the T08 and T12 plots had a loamy texture measured in 2013, and the detailed information was
provided in the online supplementary materials (Table S1). The soil pH at the time of tea seedling transplanting was 6.0 (Yao
et al., 2018). Irrigation was adopted following the typically regional management practice. Daily fluxes of $N_2O$ and NO,
topsoil (5 cm) temperature and surface soil (0−6 cm) moisture in water-filled pore space (WFPS) for each field treatment
were observed over two full years for T08-NN, T08-UN, and T08-OM (from mid-September 2012 to mid-October 2014) and
one full year for T12-NN, T12-UN, and T12-OM (from mid-September 2013 to mid-October 2014). For more detailed
information on the field experiments and observed data, refers to Yao et al. (2015, 2018) and Table S2.

## 2.2 Model modifications

In this study, the CNMM-DNDC was modified through (i) defining and applying a soil pH regulating factor ($f_{sph}$) on plant growth and (ii) adding two processes that produce $H^+$ and thus acidify soils (Miao, 2015; Pang, 2014). These modifications were made to enable the model to simulate the responses and feedbacks between tea growth and soil pH changes.

Considering that the soil pH for tea growth is optimal within 5.0−5.4 and suitable within 4.0−6.5 (Cao et al., 2009), $f_{sph}$, a dimensionless factor (0–1), is newly parameterized as a quadratic polynomial function utilizing an average soil pH of 0−20 cm ($sph_a$) as its single independent variable (Eq. 1). Based on Eq. 1, the value of $f_{sph}$ is around 1.0 when soil pH is within 5.0−5.4, and is above 0.85 when soil pH is within 4.0−6.5. However, the transient soil pH increase induced by urea hydrolysis is not considered for affecting plant growth, which can be offset due to the soil buffering effect within a few days. At each time step of simulation, the value of $sph_a$ is updated. This parameterized factor is introduced into the model to regulate photosynthesis and thus plant growth, even though the modification to the model was not yet calibrated or validated due to a lack of sufficient field observations at the selected tea fields.

$$f_{sph} = -0.089 sph_a^2 + 0.947 sph_a - 1.51 \qquad (1)$$

The two processes newly introduced into the model to simulate additional changes in the $H^+$ concentration ($\Delta[H^+]$), thus modifying soil pH, are (i) ionization of the amino acids and other organic acids (HR) in root exudates (Rxn 1) and (ii) hydrolysis of the $Al^{3+}$ from the decomposition of tea residues due to the trimming (tea leaves and young branch) or falling of old leaves (Rxn 2).

$$HR \leftrightharpoons H^+ + R^- \qquad (Rxn\ 1)$$

$$Al^{3+} + 3H_2O \leftrightharpoons Al(OH)_3 + 3H^+ \qquad (Rxn\ 2)$$

The ionization equilibrium of organic acids is formulated in Rxn 1, wherein HR represents the category of amino acids or other organic acids in root exudates. Following Eqs. 2–4, the $H^+$ concentration changes due to the ionization of these exudate-contained acids ($\Delta[H^+]_{ex}$, mol $L^{-1}$) are calculated by solving the equations (analytical method) of Eqs. 3–4, which include the parameters of average ionization equilibrium constants for amino acids ($K_{ami}$, mol $L^{-1}$) and the other organic acids ($K_{org}$, mol $L^{-1}$) in root exudates and the molar concentrations of amino acids and organic acids in the soil water ($c_{ami}$ and $c_{org}$, respectively, in mol $L^{-1}$). As the acid ionizations are thermodynamic processes, both $K_{aim}$ and $K_{org}$ vary with soil temperature ($T$, in ℃). Their values at various temperature conditions are given via the correction of their constant values at 25 ℃ for both acids, i.e., $K_{aims} = K_{orgs} = 1.75 \times 10^{-5}$ mol $L^{-1}$ (Fu, 1999), using a temperature regulating factor, $f_{acid}$ (Eqs. 5–6). The function form for parameterizing $f_{acid}$ (Eq. 7) was adapted from Li (2016). The molar concentrations of the acids and $H^+$ in the soil water are calculated using Eqs. 8–10. In these equations, $10^{-4}$ is a dimension adaptor (for each 3-hour time step), $h$ denotes the thickness of each soil layer (m), SM stands for the soil moisture in volumetric water content ($m^3\ m^{-3}$), $M_{ami}$ and $M_{org}$ represent the average molar mass of amino acids (128 g $mol^{-1}$) and the other organic acids (119 g $mol^{-1}$), respectively,

in root exudates (Fu, 1999), $a_{ami}$ and $a_{org}$ are the mass fractions of the two categories of acids in root exudates
(dimensionless), Ex is the root exudates in the soil layer (kg ha$^{-1}$) and sph$'$ denotes the soil pH, and $c_{H(soil)}$ is the H$^+$
concentration corresponding to the most lastly updated pH. At each time step (3 h) of the model simulation, 6% of the net
primary productivity is assumed to be released into the soil profile via root exudation. This assumption was made by
referring to the experimental data of some other tree species (Miao, 2015). The Ex in the soil layer is determined by
portioning the exudate quantity according to the vertical distribution of the root biomass in the soil profile of root depth.

$$\Delta[H^+]_{ex} = \Delta[H^+]_{ami} + \Delta[H^+]_{org} \tag{2}$$

$$K_{ami} = \Delta[H^+]_{ami} (c_{H(soil)} + \Delta[H^+]_{ami})(c_{ami} - \Delta[H^+]_{ami})^{-1} \tag{3}$$

$$K_{org} = \Delta[H^+]_{org} (c_{H(soil)} + \Delta[H^+]_{org})(c_{org} - \Delta[H^+]_{org})^{-1} \tag{4}$$

$$K_{ami} = K_{amis}f_{acid} \tag{5}$$

$$K_{org} = K_{orgs}f_{acid} \tag{6}$$

$$f_{acid} = 0.81 + 0.0077T \tag{7}$$

$$c_{ami} = 10^{-4}h^{-1}SM^{-1}M_{ami}^{-1}a_{ami}Ex \tag{8}$$

$$c_{org} = 10^{-4}h^{-1}SM^{-1}M_{org}^{-1}a_{org}Ex \tag{9}$$

$$c_{H(soil)} = 10^{-sph'} \tag{10}$$

According to Rxn 2, the H$^+$ concentration changes due to the hydrolysis of Al$^{3+}$ derived from decomposition of tea
plant residues ($\Delta[H^+]_{res}$) are calculated by solving the equation (numerical method by Newton iteration) of Eq. 11. In this
equation, $K_w$ ((mol L$^{-1}$)$^2$) and $K_b$ ((mol L$^{-1}$)$^3$) denote the water dissociation constant and ionization equilibrium constant of
aluminum hydroxide (Al(OH)$_3$), respectively, and $c_{Al(III)}$ is the molar concentration of Al$^{3+}$ in the soil water (mol L$^{-1}$). As
both water dissociation and Al(OH)$_3$ ionization are also thermodynamic processes, their equilibrium constants
(dimensionless) vary with soil temperature and are thus determined following Eqs. 12–13, wherein the values at 25 °C, i.e.,
$K_{ws} = 1\times10^{-14}$ (mol L$^{-1}$)$^2$ and $K_{bs} = 1.3\times10^{-33}$ (mol L$^{-1}$)$^3$ for water and Al(OH)$_3$, respectively (Fu, 1999), are corrected by the
factors $f_w$ and $f_b$, respectively. The parameterization for $f_w$ (Eq. 14) was cited from Li (2016), and $f_b$ was parameterized by Eq.
15. For calculation of $c_{Al(III)}$ in Eq. 16, $M_{Al}$ denotes the molar mass of Al$^{3+}$ (27 g mol$^{-1}$), $b$ the fraction of hydrolyzed Al(OH)$_3$
(dimensionless), $c$ the Al content in tea residues (kg kg$^{-1}$ dry matter), and Res the quantity of tea residues in dry matter (kg
ha$^{-1}$). As the Al concentration in tea leaves varied from 1.2 to 2.7 mg g$^{-1}$ dry matter, the $c$ value was set as $2.3\times10^{-3}$ kg kg$^{-1}$
dry matter (Hajiboland et al., 2015; Xu et al., 2006).

$$K_w^3K_b^{-1} = \Delta[H^+]_{res}(c_{H(soil)} + \Delta[H^+]_{res})^3(c_{Al(III)} - \Delta[H^+]_{res}/3)^{-1} \tag{11}$$

$$K_w = K_{ws}f_w \tag{12}$$

$$K_b = K_{bs}f_b \tag{13}$$

$$f_w = 0.1945e^{0.0645T} \tag{14}$$

$$f_b = 1.09 - 0.0037T \tag{15}$$

$$c_{Al(III)} = 10^{-4}h^{-1}SM^{-1}M_{Al}^{-1}bc\text{Res} \tag{16}$$

Using the $H^+$ concentration changes calculated above, the soil pH most lastly modified by the originally existing
processes, or at the last time step of simulation, is further updated by Eq. 17. The soil pH updated by Eq. 17 is used to update
the independent variable of Eq. 1 so as to provide an update of $f_{sph}$.

$$\text{sph} = -\lg(c_{H(soil)} + \Delta[H^+]_{ex} + \Delta[H^+]_{res}) \tag{17}$$

For the processes newly added above, the unknown parameters, $a_{ami}$, $a_{org}$ and $b$, were calibrated in this study using the
observed soil pH in the T08 and T12 plots. The independent variables of $T$, $h$, SM, and Res, as well as the net primary
production and the root biomass distribution in the soil profile required to calculate Ex, are provided by the model
simulations at each time step.
The soil pH dynamics affected by the urea hydrolysis, soil buffering and manure application have already been
considered in the original CNMM-DNDC (Table S3). The CNMM-DNDC with and without the above modifications is
hereafter referred to as the original and modified model, respectively.

## 2.3 Evaluation of model simulations for emissions of both gases

The model performances in simulating $N_2O$ and NO emissions from the tea plantations were evaluated by comparing
the simulations of the original and modified models with the field observations. The required input of hourly meteorological
data (air temperature, precipitation, wind speed, solar radiation, humidity) for years with gas flux measurements (2012–2014)
were obtained from the meteorological station at the field site, while those in 2008−2011 were adapted from the daily data at
the nearby government meteorological station (provided by the National Meteorological Information Center:
http://data.cma.cn/) by referring to the diurnal patterns of the hourly data observed and provided by the Shennongjia Station
(~40 km south of the tea fields) of the Chinese Ecosystem Research Network. The aforementioned observations were used
for the required inputs of soil properties (SOC, total nitrogen, mass fraction of clay, pH, and bulk density). The required
inputs of field capacity and wilting point (0.38 and 0.16, respectively, in volumetric water content) were calculated by the
pedo-transfer functions used by Li et al. (2019). According to the local survey, the initial biomass of tea seedling
transplanting was set as 1500 kg dry matter (DM) ha$^{-1}$. The harvest of buds and the trim of canopy were started at the 4[th]
year after transplanting (YAT), following the local conventional practices. The bud tea was harvested in the T08 from April
to May and August to October in the 4[th], 5[th] and 6[th] YAT, with annual yields of 37.5−150 kg DM ha$^{-1}$. The tea plants were
trimmed twice per year in June and November and nearly 40% of the aboveground biomass was cut and left on the ground.
The detailed management practices during the gas measurement period were obtained from Yao et al. (2015, 2018), which
were also adopted during the remaining period of simulation. The simulated soil profile (0−100 cm depth) was divided into
20 layers. The thickness of each layer was 1, 5 and 10 cm for the top 10, middle 2, and other 8 layers, respectively. The time
step of simulation was set as 3 hours. The measured data (Yao et al., 2015, 2018) used for evaluating the model performance
included the topsoil temperature and moisture and, the daily fluxes of $N_2O$ and NO emissions from the T08-NN, T08-UN,
and T08-OM in 2012−2014 and, the T12-NN, T12-UN, and T12-OM in 2013−2014 (Figure 1).

**2.4 Investigation of fertilization and stand age effects on emissions of both gases**

In the field cases involved in this study, the short-term replacement of urea with oilcake was implemented in the 2nd
(T12) or 5th−6th (T08) YAT following the land use change from long-term paddy rice cultivation to perennial tea plantation.
Based on the field observations of $N_2O$ and NO emissions reported by Yao et al. (2015, 2018), the performance of the
original and modified models in simulating the effects of the urea replacement by oilcake was examined through the
comparison between the model relative bias (MRB) magnitudes and the observational error indicated by the coefficient of
variation (CV). An absolute MRB (|MRB|) smaller than the two times CV of the spatially replicated observations, which
represented the observational uncertainty at the 95% confidence interval (CI), was considered to indicate a statistically
satisfactory performance (Dubache et al., 2019). For this examination, the urea replacement effects ($E_{ur}$, in %) on the $N_2O$
and NO emissions and their relative observational errors ($\varepsilon_{ur}$, in %) at the 95% CI were calculated using Eqs. 18−19. In both
equations, $\overline{E_o}$ and $\overline{E_u}$ (in kg N ha$^{-1}$ yr$^{-1}$) denote the mean annual emission of $N_2O$ or NO from the OM and UN treatments,
respectively, and $\delta_o$ and $\delta_u$ (in kg N ha$^{-1}$ yr$^{-1}$) signify the corresponding observational errors in two times standard deviation
(SD). Equation 19 is analytically established according to Eq. 18 and following the general error propagation theory. The
observed data were directly cited or adapted from Yao et al. (2015, 2018).

$$E_{ur} = 100(\overline{E_o}/\overline{E_u} - 1) \tag{18}$$

$$\varepsilon_{ur} = 100(\overline{E_u}^{-2}\delta_o^2 + \overline{E_o}^2\overline{E_u}^{-4}\delta_u^2)^{1/2}/(\overline{E_o}/\overline{E_u} - 1) \tag{19}$$

The virtual experiments were designed to evaluate the performance of the original and modified models in simulating
the annual $EF_d$s and to investigate the effects of fertilizer nitrogen doses on $EF_d$s. For each field treatment exclusively
applied with urea or oilcake in 2013 or 2014, virtual experiments against nitrogen addition rates varying from zero to 600
(with an interval of 50) kg N ha$^{-1}$ yr$^{-1}$ were carried out. For each treatment, the gradient nitrogen doses were set only in the
experimental year but remained at 450 kg N ha$^{-1}$ yr$^{-1}$ in the other year(s). The annual $EF_d$s (the fraction of the increased
fertilizer nitrogen input released in the form of $N_2O$ or NO within the one-year period after fertilization) in percentage for the
nitrogen dose gradients were simulated at each gradient with an interval ($N_{50}$) of 50 kg N ha$^{-1}$ yr$^{-1}$, following Eq. 20,
wherein $E_{50+}$ and $E_{50-}$ denote the simulated annual emissions of $N_2O$ or NO at the higher and lower fertilizer nitrogen dose of
the gradient, respectively.

$$EF_d = 100(E_{50+} - E_{50-})/N_{50} \tag{20}$$

The stand age effects on annual $N_2O$ and NO emissions in the early stage (1−6 years) or a full tea plant lifetime (35
years) of a plantation can be investigated if the applicability of the model was proven using available observations at the
field site. Acceptable model applicability can be indicated by a smaller average |MRB| than two times CV of the spatially
replicated observations. The effects of the stand ages during the early stage (1st to 6th YAT) or the full tea lifetime (usually
until approximately the 35th YAT in the region) can be investigated using a virtual experiment. The tea plantation in this
virtual experiment was purely fertilized with urea at the conventional timings and doses. Any influencing factor other than
stand age should be excluded from this virtual experiment. To ensure the simulations of all the stand ages can be driven by
the same meteorological conditions that would be the same as the measured data during the year-round period from
September 17[th], 2013 to October 9[th], 2014, 35 independent scenarios were designed. Thus, the seedling transplanting for the
stand ages of 35, 34, ..., 1 year were set to occur in March of 1979, 1980, ..., and 2013, respectively. The field management
practices for T08-UN would be set for each stand age scenario.
**2.5 Statistics and method to quantify uncertainties**
The statistical criteria used in this study to evaluate the model performance include (i) the index of agreement (IA), (ii)
the Nash-Sutcliffe index (NSI), (iii) the determination coefficient ($R^2$) and slope of the zero-intercept univariate linear
regression (ZIR) of the observations against the simulations, and (iv) the MRB. The IA falls between 0 and 1, with a value
closer to 1 indicating a better simulation. An NSI value (ranged from minus infinity to 1) between 0 and 1 shows acceptable
model performance, whereas closer to 1 is better. Better model performance is indicated by a slope and an $R^2$ value that is
closer to 1 in a significant ZIR. The performance is regarded as acceptable if a significant ZIR with its slope closer to 1 can
be obtained or the |MRB| on average is smaller than the two times CV of replicated observations. Akaike information
criterion (AIC) is applied to evaluate the significance of the multivariate linear regression. The additional independent
variable is significant when the value of AIC decreases. For more details on these criteria, refer to Eqs. S1−5 in Table S4.
The model simulation error ($\varepsilon_s$) indicated the simulated bias diverging from the observation. It represented the total
simulation uncertainty and was made of the uncertainty due to the model insufficiencies in scientific structure or process
parameters ($\varepsilon_{model}$) and that due to the uncertainties of input items ($\varepsilon_{input}$) (Zhang et al., 2019). For the investigation of stand
age effects, the mean relative $\varepsilon_s$ and its random uncertainty (95% CI) for either gas were estimated as the mean and the two
times SD of the MRBs relative to the observations for three stand ages (i.e., those in the T12-UN and T08-UN fields in the
2[nd] and 5[th]−6[th] YAT). The relative $\varepsilon_s$ values for a gas were regarded to be equal among the different stand age scenarios. The
mean or the two times SD of the relative $\varepsilon_s$ was converted to its absolute magnitude through multiplying it with the product
of an adjustment factor and the simulated gas emission quantity. The adjustment factor was obtained from the model
validation of the three stand ages, which was estimated as the mean of the ratios of individual observations to simulations.
Since the uncertainties of the model input items were known as random errors, the $\varepsilon_{input}$ was a random error. It was estimated
using the Monte Carlo test with Latin hypercube sampling (Helton and Davis, 2003) within the uncertain ranges (95% CI) of
sensitive input items, which included the soil properties (bulk density, pH, clay fraction, SOC and soil total nitrogen content)
(e.g., Li, 2016), thermal degree days (TDD) for maturity, and nitrogen content in the different plant stages (seedling, early
and harvest stages). According to the measurement errors, the uncertain ranges of the input items were 1.11−1.35 g cm[−3] for
bulk density, 5.6−6.4 for pH, 0.120−0.128 for clay fraction, 9.6−13.6 g kg[−1] for SOC content, and 1.00−1.48 g kg[−1] for total
nitrogen content. The uncertainties of the TDD and, plant nitrogen content in the three stages were assumed to be ±5% of
the default values, which were 2500 ℃, and 7.8, 6.8 and 6.0 g N kg[−1] DM, respectively. A uniform distribution for sampling
was assumed in the Monte Carlo test, in which the simulations were iterated until the mean of the simulated gas emission
quantities for all iterations converged to certain level within the tolerance of 1%. The $\varepsilon_{input}$ at the 95% CI was presented as
the double SDs of these iterated simulations.
If not specified, errors are presented hereafter at the 95% CI.
In this study, the statistical analyses and graphical comparisons were performed with the SPSS Statistics Client 19.0
(SPSS Inc., Chicago, USA) and Origin 8.0 (OriginLab, Northampton, MA, USA) software packages.

## 3 Results

### 3.1 Calibration of modified model for soil pH simulation

Using the topsoil (0−15 cm) pH (6.0) prior to tea seedling transplanting and the values of 5.4 and 5.0 measured in T12-
UN and T08-UN, respectively, in September 2013, each of the three parameters involved in the modified model (Eqs. 8−9
and 16) was calibrated to $5.0 \times 10^{-4}$ for $a_{ami}$ and $a_{org}$, and $1.0 \times 10^{-3}$ for $b$, respectively. The simulations of the modified
CNMM-DNDC with these calibrated parameters resulted in topsoil (0−15 cm) pH values of 5.42 and 5.01 in the T12-UN
and T08-UN fields, respectively, in September 2013, which were consistent with the observations. Differently, the soil pH
simulated by the original model remained nearly constant (approximately 6.0) during the 6-year period, despite the transient
increases due to urea hydrolysis. Nevertheless, it is still required to validate the simulations of the modified model on the soil
pH changes due to tea growth using more field observations under different conditions.

### 3.2 Model validation for soil environment and emissions of both gases

Both the original and modified models accurately predicted the seasonal dynamics and magnitudes of topsoil
temperature and moisture (Figures 1a−b). The satisfactory model performance was indicated by the statistics in Table 1.
The measured daily $N_2O$ and NO fluxes were highly variable across the entire observation period (Figures 1c−n). The
original and modified models generally captured the seasonal patterns of both gases for different field treatments, even
though the magnitudes of some peak fluxes were inconsistent with the observations. In comparison, the original model
generally overestimated the peak emissions of both gases. The performance of both models was similar and satisfying for the
daily fluxes as indicated by the comparably IA, NSI, and ZIR slope and $R^2$ values (Table 1). For the original model, three
(NO) and five ($N_2O$) of the nine individual simulations for each gas showed |MRBs| larger than the corresponding observed
two times CV, while |MRBs| larger than the observed two times CV were four (NO and $N_2O$) for the modified model (Table
S5). However, the statistics of both models still indicated agreements for annual emissions, with the IA and NSI values of
0.96−0.98 and 0.81−0.94, respectively, for $N_2O$ and NO (Table 1). In addition, the modified model improved the simulation
of annual $N_2O$ emission, with higher IA, NSI, ZIR slope and $R^2$ values of 0.98, 0.94, 0.97 and 0.94 ($p < 0.001$), respectively
(Table 1, Figure 2). These results indicate that the modified CNMM-DNDC can effectively simulate the daily and annual
emissions of both gases from the tested tea plantations. Additionally, the modified model resulted in adjustment factors of
0.86 and 1.09 and relative $\varepsilon_s$ values of $17 \pm 20\%$ and $-8 \pm 14\%$ for the annual $N_2O$ and NO emissions from the UN
treatments and adjustment factors of 1.00 and 0.97 and relative $\varepsilon_s$ values of $0.2 \pm 24\%$ and $6 \pm 38\%$ for the $N_2O$ and NO
emissions from the OM plots, respectively. These adjustment factors and relative $\varepsilon_s$ were used to estimate the absolute total
errors of the simulated emissions.

### 3.3 Effects of organic fertilization on emissions of both gases

According to the field observations, the short-term replacement of urea by oilcake stimulated the annual $N_2O$
emissions by ~62% (ranging between $35-95\%$ or $5.3-13.7$ kg N ha$^{-1}$ yr$^{-1}$) but simultaneously mitigated the annual NO
emissions by ~25% (ranging between $12-33\%$ or $2.4-6.0$ kg N ha$^{-1}$ yr$^{-1}$). Based on the statistical analysis using linear mixed
models, both the stimulation and mitigation effects were significant ($p < 0.05$) (Yao et al., 2015). The average relative
observational errors of these effects were ~97% (ranging between $92-106\%$) for $N_2O$ and ~73% (ranging between $60-83\%$)
for NO (adapted from Yao et al., 2015, 2018; Table S6). The simulated effects of the fertilizer replacement on annual $N_2O$
emissions by the modified model showed stimulations by ~36% (ranging between $24-49\%$ or $5.7-9.1$ kg N ha$^{-1}$ yr$^{-1}$), with
|MRB| of ~36% (ranging between $4-56\%$) (Table S6). The |MRB| magnitudes were significantly lower than the relative
observational errors ($p = 0.02$), indicating consistency between the simulated and observed effects. The inhibition effects of
the fertilizer replacement on annual NO emissions were about 14% (varying between $1-21\%$ or $0.1-4.1$ kg N ha$^{-1}$ yr$^{-1}$) by
the modified model except for some underestimation, which indicated the consistent effects between the simulations and
observations (Table S6). As these results suggest, the model with improvements in scientific processes could simulate the
effects of short-term replacement of urea by oilcake on $N_2O$ and NO emissions in the early stage of the new tea plantations.

### 3.4 Nitrogen dose effects on annual direct emission factors of both gases

As Figures 3a−b and S1a−b show, the simulated annual emissions of either gas non-linearly varied with the nitrogen
addition rate in form of urea or oilcake. Accordingly, for the modified model, the simulated annual $EF_ds$ of either gas at
different levels of fertilizer doses increased linearly with the urea addition rates (Figures 3c−d) but nonlinearly with the
organic manure addition rates (Figures 3e−f). In comparison with the linear fittings for the manure treatment, the
relationships were better fitted the non-linear curves, as indicated by the decreased AIC values (1.74 versus 1.72 for $N_2O$ and
0.53 versus 0.31 for NO). The simulations by the original model showed similar results with those of the modified model
(Figures 3c−f and S1c−f). The original and modified model simulations of annual gas emissions for the two experimental
nitrogen doses (zero and 450 kg N ha$^{-1}$ yr$^{-1}$) resulted in significantly consistent $EF_ds$ with the field observations for $N_2O$
(Figure 4a). In comparison with the original model, the modified model performed better in simulating the $EF_ds$ of $N_2O$,
increasing IA from 0.78 to 0.89 and NSI from 0.10 to 0.64 (Table 1). For NO, the simulated annual $EF_ds$ by both models
tended to be positively related with the field observations (Figure 4b), with acceptable IA of $0.85-0.89$ and NSI of $0.38-0.50$
(Table 1). These results imply that, compared with the original model, the modified version with the pH reduction processes
added in this study could be applied to simulate the $EF_ds$ of either gas from tea plantations under different field conditions.

## 3.5 Effects of stand ages on emissions of both gases

The measured annual $N_2O$ and NO emissions from the T12-UN and T08-UN fields in the $2^{nd}$ and $5^{th}–6^{th}$ year ranged from 14.4−21.1 and 13.1−19.4 kg N $ha^{-1}$ $yr^{-1}$, with double CVs of ~43% (ranging from 9−72%) and ~13% (ranging from 6−21%), respectively (Yao et al., 2015, 2018). The original model simulations of annual $N_2O$ and NO emissions showed |MRB| of ~33% (ranging from 6−76%) and ~6% (ranging from 3−10%) respectively, while |MRBs| of the annual $N_2O$ and NO emissions were ~17% (ranging from 11−28%) and ~8% (ranging from 1−14%) for the modified model. The |MRB| on average for either gas (by both models) was smaller than the two times CV on average in the observations. This evaluation indicates that the modified model with the new processes could also reliably simulate the emissions of both gases under different stand ages and therefore be applicable for investigating stand age effects in long time using a virtual experiment.

For the modified model, the simulated daily topsoil1 (0−15 cm) pH during the early 6-year period basally declined gradually, with a temporary sudden pulse immediately following the urea application events either in spring or autumn (Figure 5a). Although the simulated pH declined from the initial value of 6.0 to less than 5.0, it was still higher than 4.5 which was the threshold set in the model to trigger the chemo-denitrification process. Different from the slightly nonlinear changes in the simulated basal pH, the simulated annual emissions of $N_2O$ and NO gradually increased with the stand ages in the early four or five years, but then decreased gradually. The variation trend for the simulated annual emissions of either gas against the early stand ages (1−6 years) could be fitted by a quadratic polynomial equation instead of the linear relationship as indicated by the decreased AIC values for the non-linear fitting as compared with that for linear regression (−1.75 versus 0.66 for $N_2O$, and −3.67 versus 0.55 for NO). Similar nonlinear relationships were also obtained for the simulations by the original model (Figure S2). As Figure 5 indicated, almost all the field observations in the fertilized fields not only generally fell within the range of the uncertainty induced by the input items, but also within the upper and lower bounds of uncertainty (95% CI) of the regressions. Compared with the uncertainty induced by the inputs ($\varepsilon_{input}$), the absolute values of the total model uncertainty ($\varepsilon_s$) were much smaller, which only accounted for 32% and 35% of the $\varepsilon_{input}$ for $N_2O$ and NO, respectively.

Although the performances of both models in simulating the emissions of both gases were comparable in early stand ages, the original and modified model thereafter performed quite differently. The 35-year simulations demonstrated that the above polynomial functions derived from the original model simulation applied for both gases during the full tea lifetime; but those derived from the modified model did not apply for the mid to late stand stages (Figure 6a). After the annual emissions of both gases simulated by the modified model reached peak values, they decreased near-linearly until around the $15^{th}$ YAT, when the chemo-denitrification process was triggered by the pH threshold (4.5) set in the model. Thereafter, the emission of either gas gradually increased at a very small annual increment (Figure 6a). Thus, the emissions of both gases simulated by the original model were about two times those by the modified model during the mid to late tea stand ages. The $\varepsilon_s$ of the simulation by the modified model were ranged from 2.11 to 2.89 and -1.63 to -0.78 for $N_2O$ and NO, respectively (Figure 6a), indicating the potential overestimation or underestimation of either gas for 35-year simulations. Meanwhile, different from the stable topsoil pH (except for the sudden pulse due to urea hydrolysis) by the original model, the simulated

basal pH of 0−15 cm by the modified model continued to decrease, finally reaching 3.74 (Figures 6b−c). In addition, the 35-
year simulation showed that the negative effects of soil pH on tea yield increased with the stand ages, resulting in the
reduction by 0.3−3.4% (Figure 6d). These results suggest that the modifications by adding the processes regulating soil pH
dynamics are necessary for accurately quantifying the long-term emissions of $N_2O$ and NO from tea plantations.
**4 Discussion**
**4.1 Model modifications**
The modified CNMM-DNDC was hypothesized to reflect the general knowledge that tea can grow in soils with a
suitable pH within 4.0−6.5 (Cao et al., 2009). But the transient increase of soil pH due to urea hydrolysis has no impact on
plant growth, as the soil pH could be recovered within a few days due to soil buffering effect. Due to the lack of observed tea
yields, the parameterized impact of soil pH on tea growth could not be calibrated or validated in this study, but virtual
experiments showed increased yield reduction with increasing stand age, implicating the intensified negative effects on plant
growth for older tea plantations. The newly added scientific processes relating to pH reduction were calibrated using the
observed soil pH for different stand ages during the early stage of a tea plantation. Although the simulations showed that the
modified CNMM-DNDC with the calibrated parameters could accurately reflect the basal soil pH declination during the
early years, validation was still missing due to a lack of available independent observation of pH. However, the studies of the
tea plantations in Jiangsu and Anhui provinces showed that the average soil pH (0−20 cm) declination rate was 0.06 pH yr$^{-1}$
(Luo, 2006; Su, 2018). For the simulation of 35-year tea plantation in this study, the calculated average annual soil (0−20 cm)
pH declination rate was close to the reports with the value of 0.064. Therefore, the consistent declination rate indicates the
modifications improve the scientific mechanisms of the biogeochemical model which could be applied for long time
simulation. As the actual soil pH would not decline constantly (Yao et al., 2018), the validation of soil pH dynamics for long
time is still necessary. The simulated annual emissions by both models were comparable in the early tea stand ages, but those
by the modified model were much lower in the mid to late stages of tea lifetime. According to the modifications, the
different annual emissions of both gases should be primarily attributed to the soil pH differences. Therefore, the proper
simulation of soil pH declination for long time increased the reliability of the simulated variation of annual emissions even
though validation of the differences was still missing due to lacking of field observations. Thus, further study is still needed
to confirm the general model applicability, especially for the simulations of long term yields, soil pH dynamics, $N_2O$ and NO
emissions from tea plantations subject to different conditions.
**4.2 Model performance**
This study was the first study testing the original or modified models against the measurements of $N_2O$ and NO
emissions from a tea plantation. The results showed that both the original and modified models accurately captured the high
temporal variations of daily $N_2O$ and NO emissions driven by the application of fertilizers, stand ages and weather
conditions (Yao et al., 2015, 2018). Many previous studies did not report the $R^2$ of regressions between the observed and
simulated daily fluxes of either gas, usually due to poor model performance (Bell et al., 2012; Bouwman et al., 2010;
Butterbach-Bahl et al., 2009). Considering the large uncertainties of field measurements as indicated by the SDs of the
observations and the complexity of the management practices, the performance of the modified model for either gas was
encouraging. Yao et al. (2015, 2018) obtained significant revised "hole-in-pipe" (HIP) regressions for the observed daily
$N_2O$ plus NO fluxes as the dependent variable and the soil ammonium plus nitrate concentrations, temperature and moisture
as the multiple independent variables. Compared with the $R^2$ values of the original HIP regressions fitting the daily
observations, those of the revised HIP model more than doubled and were up to 0.95−0.97 (Yao et al., 2015, 2018).
Similarly, the daily simulations by the modified model also resulted in significant revised HIP regressions that showed more
than doubled $R^2$ (0.48−0.55) in comparison with the values (0.01−0.12) of the original HIP (Mei et al., 2011), despite of the
smaller determination coefficients than those for the field observations. The improvements of the revised HIP regression by
both observations and simulations were due to the consideration of the temperature- and moisture-regulated effects of
nitrogen substrates for both nitrification and denitrification processes that produce $N_2O$ and NO.

For the annual $N_2O$ emissions, the statistics of the modified model were all better than the original model, indicating
the modifications about soil pH reduction improve the model performance in tea plantations. Thus, the simulated
corresponding effects of organic fertilization and $EF_ds$ by the modified model were more consistent with the observations.
However, the simulated annual NO emissions by the modified model were not much improved in comparison with those by
the original model. The underestimation (2.56 kg N $ha^{-1}$ $yr^{-1}$) and overestimation (3.29 kg N $ha^{-1}$ $yr^{-1}$) of the NO emission in
2014 for T08-UN and T08-OM, respectively, resulted in the significant underestimation of the inhibition effects and
increased model relative bias for the modified model. The inhibited NO emissions were also partly attributed to the soil
heterotrophic nitrification (Yao et al., 2015), which is the direct oxidation of organic nitrogen to nitrate without passing
through mineralization. However, the heterotrophic nitrification was not considered in the model, which may result in the
overestimated NO emissions in 2014 for the manure treatments by both models. In addition, compared with the original
model, the underestimated NO emission mentioned above was also the key reason for the unsatisfactory simulation of $EF_ds$,
which led to the increment of the ZIR slope by 8% (1.0 for the ZIR without T08-UN and 1.08 for the ZIR with T08-UN).
Therefore, further study is still required for validating the model performance in simulating NO emissions under different
fertilization conditions.
**4.3 Contribution of the dominant process for emissions of both gases**

The CNMM-DNDC model simulates the emissions of $N_2O$ and NO from nitrification and denitrification separately,
and then sums them up to give the overall emissions of either gas contributed by both processes (e.g., Li, 2016; Zhang et al.,
2018). Some researchers have used the NO and $N_2O$ molar ratio levels higher or lower than 1 to indicate nitrification or
denitrification as the dominant process for the emissions of either gas (e.g., Yamulki et al., 1995). However, Wang et al.
(2013) have indicated that such criteria may not be applicable, as they commonly observed molar ratios greater than 1 under
strict anaerobic conditions with low to moderate initial nitrate concentrations in a calcareous soil. This viewpoint could be
supported by the simulated major contributions of the denitrification process by both models, accounting for 63−67% and
59−62% of the annual $N_2O$ and NO emissions, respectively, for all the fertilized fields. These larger contributions from the
denitrification process could be at least partially attributed to the hot and humid climate from April to September, which
resulted in favourable soil moisture and thus facilitated the $N_2O$ and NO emissions. This explanation could be supported by
the simulated soil moisture and $N_2O$ emissions from the T08-UN treatment with observations in two consecutive full years.
The simulated daily soil moisture falling in the range of 60−90% WFPS appeared at a frequency of only 40% during the
two-year period. However, the simulated cumulative $N_2O$ emissions (25.7 kg N ha$^{-1}$) occurring on the days with such
relatively high moisture contents accounted for 61% of the total modelled quantity of this gas (42.0 kg N ha$^{-1}$). It is accepted
that nitrification generally dominates $N_2O$ production in soils with less than 60% WFPS (e.g., Chen et al., 2013). The
dominant contributions of denitrification to $N_2O$ and NO emissions by the simulations could also be supported by previous
experimental/modelling studies (Chen et al., 2017; Zhang et al., 2017). However, direct validation of the simulations by the
original/modified model on the contributions of nitrification or denitrification is still lacking, due to no available direct
measurement of $N_2O$ or NO emissions from either process. This challenge will need to be overcome in future studies.
**4.4 Effects of organic fertilization on emissions of both gases**
For the tea plantations, the applied fertilizers and the retained nitrogen in the soil are consumed by plant uptake,
microbial processes and physical losses through ammonia volatilization and nitrate leaching (e.g., Zhang et al., 2015).
Accordingly, changes in fertilizer types would affect the nitrogen transformation from the fertilizer to those available for the
losses, thus altering the $N_2O$ and NO emissions (e.g., Deng et al., 2013; Goulding et al., 2008; Skinner et al., 2014). Organic
fertilization has been widely encouraged in tea cultivation since it can reduce synthetic nitrogen inputs into the biosphere
while improving both soil fertility and carbon sequestration (e.g., Skinner et al., 2014; Liang et al., 2011; Meng et al., 2005).
Yao et al. (2015) observed that short-term replacement of urea with oilcake, which is characterized by a low carbon to
nitrogen ratio, stimulated $N_2O$ emissions to a large extent while inhibiting NO releases to a relatively small extent. These
observed effects were generally simulated by the original and modified CNMM-DNDC, especially the increased $N_2O$
emissions.
According to the model simulations, the stimulated $N_2O$ emissions were jointly attributed to (i) the enhanced
production of this gas, as well as nitrate, in promoted nitrification and (ii) the enhanced production of this gas in promoted
denitrification. The promoted nitrification was due to less ammonia volatilization derived from the organic nitrogen
mineralization than the urea hydrolysis (~1.0 versus 13 kg N ha$^{-1}$ yr$^{-1}$). The oilcake mineralization slowly produced
ammonium, while the deep placement of the fertilizer also inhibited ammonia volatilization. In comparison, the urea
hydrolysis quickly transformed the fertilizer nitrogen form into ammonium within a few days following the fertilization
event, when the hydrolysis-derived pulse increase of soil pH (Figure 5a) stimulated ammonium loss by ammonia
volatilization. The denitrification was promoted not only by the improved supply of nitrate (as the primary nitrogen substrate)
from the promoted nitrification (Figure S3), but also by the enhanced activity of denitrifiers that have a very high affinity for
the carbon substrates provided by the organic manure decomposition (e.g., Li et al., 2005; Skinner et al., 2014; Snyder et al.,
2009). For the annual NO emissions of the three paired OM-versus-UN cases, the modified model resulted in consistent
decrease (1−21%) due to the full urea replacement by oilcake. The simulations showed that 0−44% of the decreases was
ascribed to the promoted nitrification (Table S5), whereby more nitrate was produced as the final product but less NO was
produced as the by-product. The remaining 56−100% of the decreases, however, was attributed to the promoted
denitrification (Table S5), whereby more NO was reduced to $N_2O$ (e.g., Meijide et al., 2007; Snyder et al., 2009; Vallejo et
al., 2006). Regarding the contributions of denitrification to the overall $N_2O$ or NO emissions, the simulations showed no
significant effect from the full urea replacement by oilcake. However, validation of this simulated insignificance is still
lacking, because no direct observations for the process contributions are currently available. Further study is still needed to
validate the model's performance in simulating the contributions of nitrification or denitrification to the emissions of either
gas from tea plantations.

**4.5 Effects of nitrogen fertilizer doses on direct emission factors of both gases**

Validation of the linear or nonlinear relationships for the urea or manure treatments from the virtual experiment was
still lacking, since there was no available data from the experimental field site for the multiple fertilizer gradients.
Nevertheless, the relationships of the simulated $EF_d$s against the nitrogen doses suggested that paired field observations of
fertilized and unfertilized treatments, or those of two largely different nitrogen addition rates, as used in many field studies
(e.g., Yao et al., 2015, 2018), would yield greatly biased $EF_d$s of either gas from the tea plantations, particularly creating a
gross underestimation for moderate to high nitrogen addition rates. This conjecture from the virtual experiment was
supported by two studies so far available for field observations of $N_2O$ emissions from tea plantations treated with nitrogen
dose gradients (Han et al., 2013 and Hou et al., 2015), even though similar literature support for NO was still lacking. These
experimental studies showed that the $EF_d$ determined by the lowest nitrogen addition rates showed 30% underestimation on
average as compared with the value by the highest nitrogen inputs (adapted from Han et al., 2013 and Hou et al., 2015).
Obviously, this study implicates the potential capacity of the modified CNMM-DNDC as a robust tool to generate $EF_d$s of
tea plantations subject to different conditions, although it is still necessary to widely validate the simulated $EF_d$s using field
observations against multiple gradients of nitrogen fertilizer doses.

**4.6 Effects of stand age on emissions of both gases**

Relative to the $N_2O$ and NO emissions in the 2[nd] or 6[th] YAT, more intensive emissions of both gases were observed in
the 5[th] YAT (Yao et al., 2015, 2018). These relatively intensified emissions were thought to result from the comprehensive
effects of increased soil nitrogen and carbon availability for nitrification and denitrification as well as reduced soil pH (Yao
et al., 2018). For either gas, the observations in the tea fields either purely applied with urea or oilcake most likely implied a

non-linear trend against stand ages with the inter-annual maximum appearing between the 2nd and 5th YAT. This implication was supported by the modified model simulations for a conventionally managed plantation over the full lifetime of tea plant, in which the inter-annual maximum of $N_2O$ emission appeared in the 4th YAT when the initial harvest of tea bud and the first canopy trim occurred. The increases in the early years were mainly ascribed to the increasing root exudates and less-woody residues returning to soil promoted by the tea plant growth. The simulated inter-annual maximum emission of $N_2O$ appeared in the year when basal soil pH reached the threshold of about 5.0. The inhibition effect of pH on microbial growth is intensified when soil pH is less than this threshold (Figure S4). The adopted pH-influencing mechanisms in the model mainly induced the diminished annual emissions of $N_2O$ following the appearance of the peak, because the emissions of $N_2O$ were associated with the microbial production. In addition to the reduced microbial activity due to low pH inhibition, the post-maximum declines in the annual gas emissions against the stand ages were also attributed to the reduced availabilities of nitrogen substrates for the microbial processes, due to (i) the higher nitrogen demand for the tea growth stimulated by the multiple bud harvests and two trims per year, as well as (ii) the too slow decomposition of woody residues for old tea remaining on the ground surface. However, experimental support is far less sufficient for these explanations on the variations of gas emissions against the stand ages in the early stage or during the full lifetime of tea growth, and thus further studies are still required. In addition, the smaller total model uncertainty, which only accounts for 33% of the uncertainty induced by inputs, indicated that increasing the reliability of the inputs of soil properties and plant growth parameters can improve the model efficiency.

The declination of emissions following the peaks of both gases may not continue throughout the entire lifetime of tea growth, as the process of chemo-denitrification would be triggered once the soil pH decreases to 4.5 or lower, thus promoting the emissions of either gas (e.g., Li, 2016; Pilegaard, 2013). Such a conjecture was well supported by the virtual experiment in this study, which demonstrated that the average soil pH (0−15 cm) decreased to the threshold in the 15th YAT and continued to decrease thereafter. In the model, the chemo-denitrification process occurring under the low soil pH ($\leq 4.5$) is assumed to transform a portion of the NO produced in the microbial nitrification and denitrification processes into $N_2O$. Before the chemo-denitrification was triggered, the simulated microbial nitrification stably accounted for ~36% of the overall $N_2O$ and ~41% of the overall NO emissions. When the chemo-denitrification occurred, its contributions to the overall simulated $N_2O$ emissions increased from ~4% to ~8% with increasing stand ages, while the microbial nitrification and denitrification accounted for ~34% and ~59%, respectively. However, these results of gas emissions from the virtual experiment still require validation with field experiments in future studies.

**5 Conclusions**

To fill a gap in the process-oriented biogeochemical model, the Catchment Nutrient Management Model - DeNitrification-DeComposition (CNMM-DNDC), in this study the effects of soil pH on tea growth and the processes that may induce soil pH reduction due to root exudation and residue decomposition during tea growth were added in the model.

Using the two-year field measurements in tea plantations at a subtropical site in central China, the original and modified models were evaluated for simulating nitrous oxide ($N_2O$) and nitric oxide (NO) emissions from this important type of agricultural ecosystem. Both the original and modified models showed comparable performance for simulating the daily and annual emissions of $N_2O$ and NO from the tested tea plantations at the early stage, especially before the initial tea harvest and the first trims. The modified model was further tested through simulating the emissions of both gases affected by short-term replacement of synthetic fertilizer (urea) with organic manure (oilcake), gradient nitrogen doses of the two fertilizers and different stand ages of new tea plantations. Both observations and simulations demonstrated that short-term replacement of urea with oilcake can largely stimulate $N_2O$ emissions and mitigate NO emissions. The simulations by the modified model also showed linear relationships between the direct emission factors ($EF_d$s) of either gas against the nitrogen doses for tea plantations amended with synthetic fertilizer and non-linear relationship for those plantations applied with organic manure. These relationships supported the hypothesis that paired field observations against two largely different nitrogen addition rates, which have been very often implemented in field studies, lead to significant biases for the measured $EF_d$s of either gas from the tea plantations. These biases particularly induce significant underestimations for the moderate to high nitrogen doses that are typically applied by farms. The model simulations also showed that annual emissions of either gas increase with stand ages within the early stage of a new tea plantation and then gradually decrease until they slightly increase again due to chemo-denitrification triggered by soil pH lower than 4.5. In conclusion, the modified CNMM-DNDC can reflect the comprehensive influences of weather, soil conditions, plant nitrogen demands, and field management practices, thus showing potential to be a powerful tool for investigating long-term emissions of $N_2O$ and NO from tea plantations under specific field management alternatives at the site or regional scale. Nevertheless, experimental data are yet too scarce to validate the model simulations of long-term soil pH changes and their effects on the emissions and $EF_d$s of both gases from tea plantations. To improve the robustness of the model for application in various tea plantations, comprehensive validations using simultaneous field observations are still necessary. The validations should not only include the variables involved in this study but also others, such as the emissions of other greenhouse gases (carbon dioxide and methane), volatilization of ammonia, hydrological nitrogen losses by leaching and surface runoff, and temporal changes in the SOC stock, which are urgently required.

**Code/Data availability**

The model, input and output databases can be obtained from the first author and all the observed data sets used in this study can be available from the co-authors.

## Author contributions

Zheng, X. and Zhang, W. contributed to develop the idea and enhance the science of this study. Zhang, W. improved the scientific processes of the model, designed and implemented the model simulations and virtual experiments and prepared the manuscript with contributions from all co-authors. Yao, Z., Liu, C., Wang, R. and Wang, K. designed and carried out the field experiments. Li, S. and Han, S. collected and established the input database for modelling. Zuo, Q. and Shi, J. provided the climate data observed in the field site.

## Competing interests

The authors declare that they have no conflict of interest.

## Acknowledgement

This study was jointly supported by the National Key R&D Program of China (2016YFD0800103), the National Natural Science Foundation of China (41603075, 40711130636) and the Chinese Academy of Sciences (ZDBS-LY-DQC007).

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

Table 1 Statistical evaluation of the original (Ori) and modified (Mod) simulations on the soil temperature (ST), soil moisture (SM), nitrous oxide
($N_2O$) and nitric oxide (NO) fluxes as daily means, annual emissions, and annual direct emission factor.

| Variable | $n$ | Mean observed | Mean simulated Ori | Mean simulated Mod | IA Ori | IA Mod | NSI Ori | NSI Mod | Slope Ori | Slope Mod | $R^2$ Ori | $R^2$ Mod |
|---|---|---|---|---|---|---|---|---|---|---|---|---|
| ST | 756 | 14.8 | 16.4 | 16.4 | 0.98 | 0.98 | 0.92 | 0.92 | 0.91 | 0.91 | 0.95 | 0.95 |
| SM | 504 | 0.51 | 0.54 | 0.54 | 0.71 | 0.71 | −0.27 | −0.27 | 0.93 | 0.93 | – | – |
| Daily $N_2O$ | 1107 | 49.5 | 46.9 | 42.9 | 0.82 | 0.80 | 0.10 | 0.18 | 0.58 | 0.63 | 0.51 | 0.42 |
| Daily NO | 1107 | 31.2 | 29.8 | 27.4 | 0.84 | 0.80 | 0.32 | 0.33 | 0.68 | 0.74 | 0.51 | 0.41 |
| Annual$N_2O$ | 9 | 16.1 | 17.8 | 16.3 | 0.96 | 0.98 | 0.81 | 0.94 | 0.88 | 0.97 | 0.87 | 0.94 |
| Annual NO | 9 | 10.3 | 11.3 | 10.4 | 0.98 | 0.98 | 0.92 | 0.94 | 0.94 | 1.02 | 0.93 | 0.94 |
| $EF_d$s of $N_2O$ | 6 | 3.99 | 5.03 | 4.65 | 0.78 | 0.89 | 0.10 | 0.64 | 0.81 | 0.88 | 0.63 | 0.80 |
| $EF_d$s of NO | 6 | 3.05 | 2.95 | 2.77 | 0.89 | 0.85 | 0.50 | 0.38 | 1.01 | 1.08 | 0.50 | 0.51 |

$n$, number of data pairs. For definitions of IA, NSI, Slope and $R^2$, refer to Subsection 2.5 in the text.

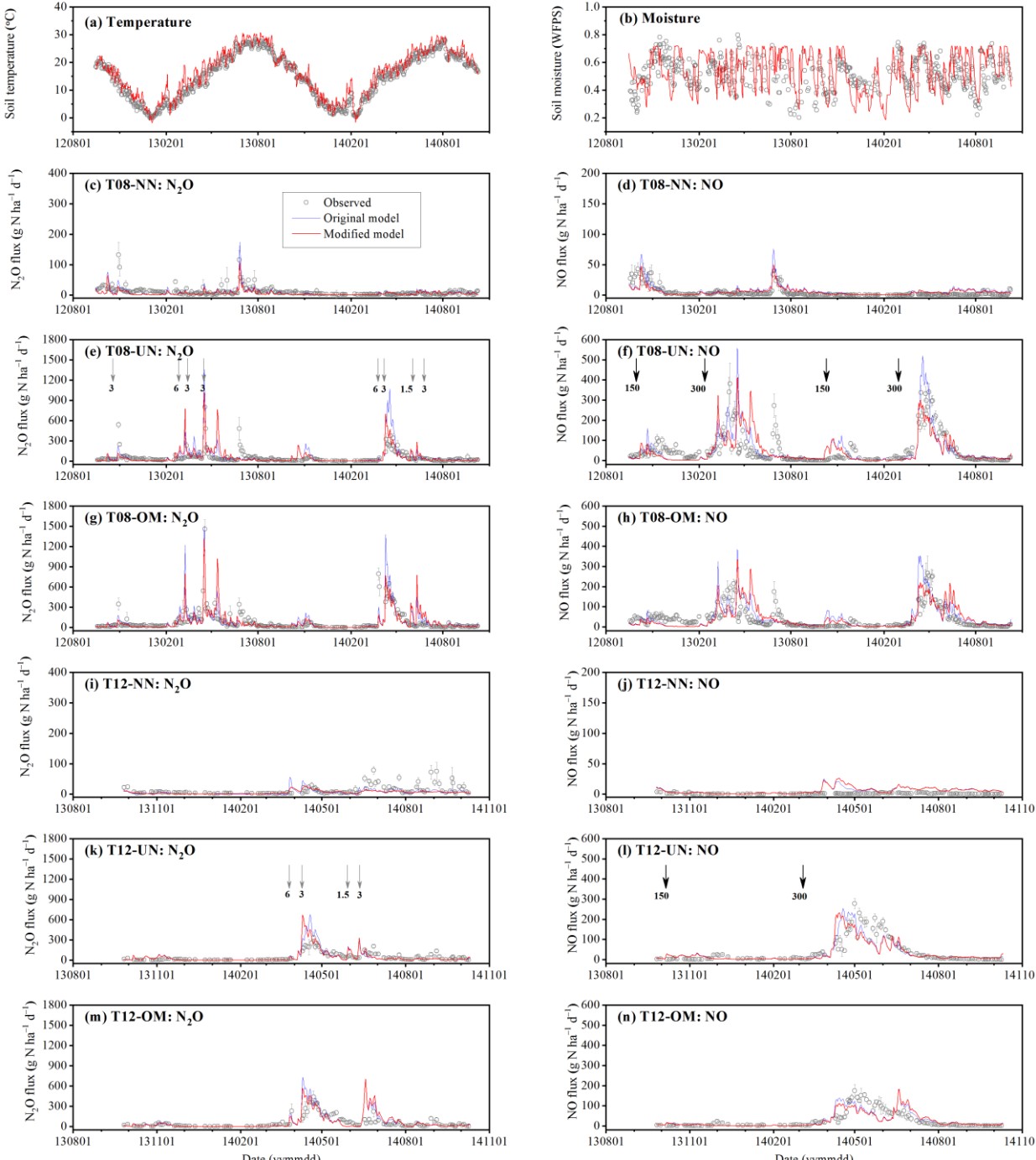


**Figure 1: Observed and simulated daily mean soil (5 cm) temperature, soil (0−6 cm) moisture, nitrous oxide (N$_2$O) and nitric oxide**
**(NO) fluxes from tea fields of different treatments by the original and modified models. T08 and T12 represent the fields with tea**
**seedling transplanting in 2008 and 2012, respectively. NN, UN and OM encode the no nitrogen applied, and fertilization with urea**
**and oilcake, respectively. The grey- and black-line arrows indicate the dates of irrigation and fertilization, respectively. The**
**number stands for the applied water amount in cm or fertilizer dose in kg N ha$^{-1}$. The vertical bar for each observation in panels**
**c−n indicates the standard error of four spatial replicates. The legends in panel c apply for all panels.**

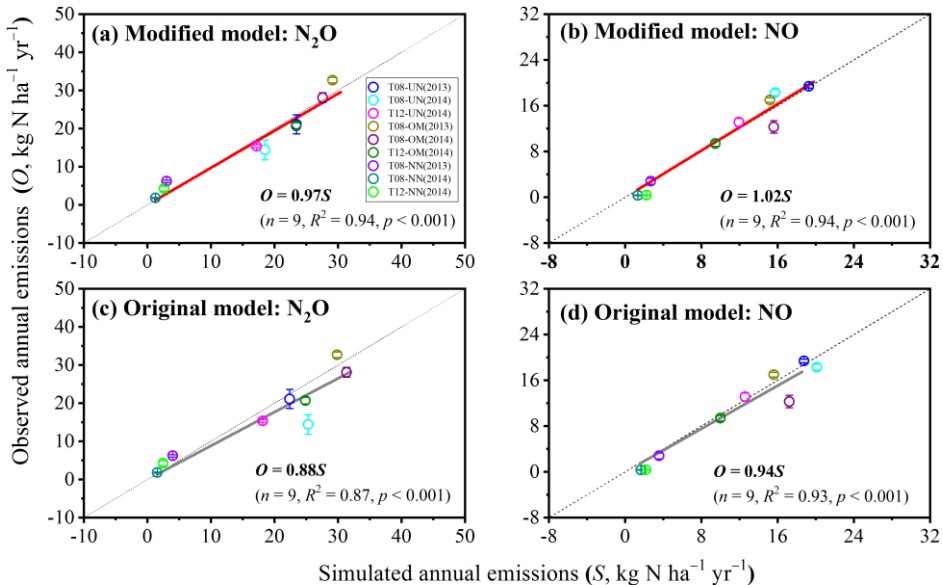


**Figure 2: Comparison between the observations and simulations of annual nitrous oxide (N$_2$O) and nitric oxide (NO) emissions.**
**The simulations were provided by the original and modified models. The red or gray solid lines illustrate the zero-intercept**
**univariate linear regressions. The vertical bars indicate the standard error of four spatial replicates. The legends in panel a apply**
**for all panels.**

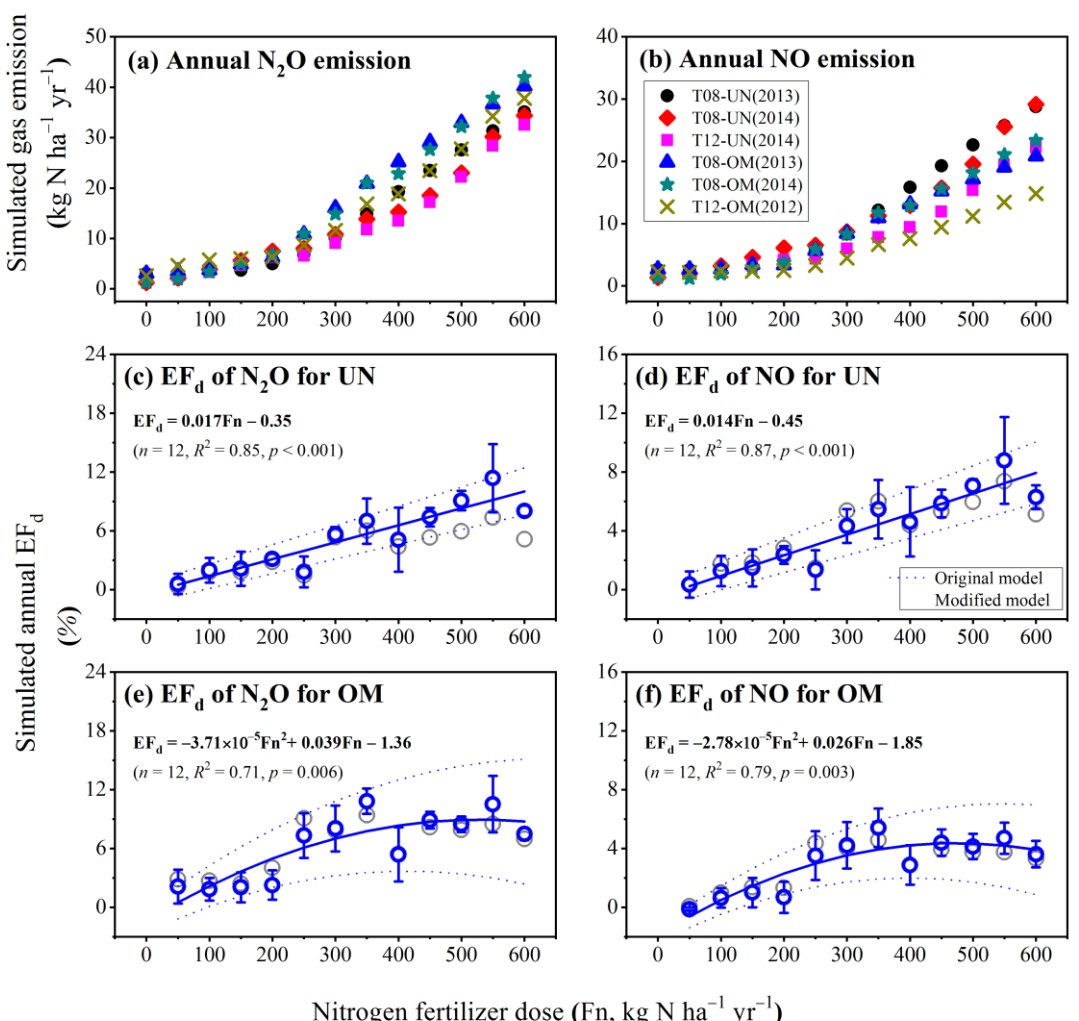


**Figure 3:** Simulated annual emission and direct emission factor ($EF_d$) of nitrous oxide ($N_2O$) and nitric oxide (NO) from tea plantations with early stand ages against nitrogen fertilizer doses. Data displayed in panels a−b were simulated by the modified model, and those in panels c−f by the original (grey circle) and modified (blue circle) models. The legends in panel b also apply for panel a, wherein T08 and T12 represent the plantations transplanted with seedlings in 2008 and 2012, respectively, UN and OM indicate the fields consecutively applied with urea since tea planting and short-term replacement of urea with oilcake, respectively, and 2013 and 2014 are the years with field observations of gas emissions. Each vertical bar in panel c−f is the standard deviation of the $EF_d$s for T08 in 2013 and 2014 and for T12 in 2013. Dashed lines are the lower and upper uncertain bounds at the 95% confidence interval for regression curves. The legends in panel d also apply for panels c, e and f.

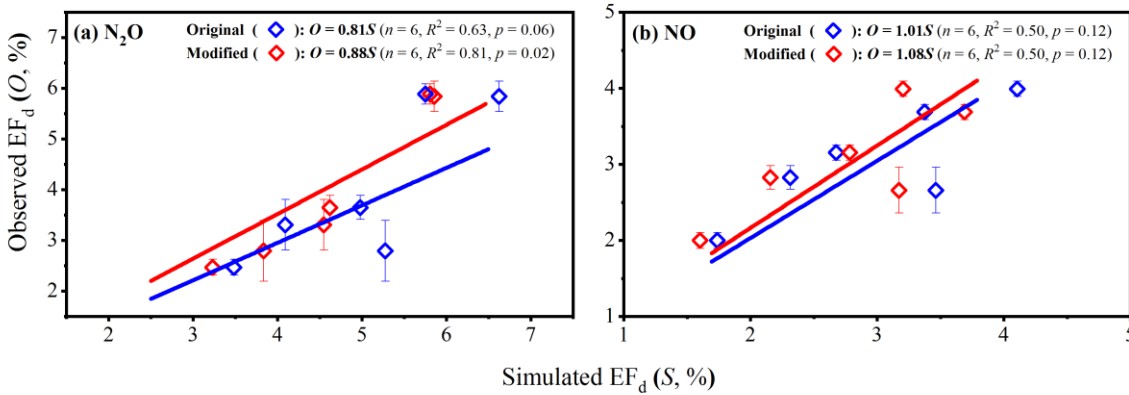


**Figure 4: Comparison between observed and simulated annual direct emission factor (EF$_d$) of nitrous oxide (N$_2$O) and nitric oxide**
**(NO) by the original and modified models from tea plantations. The vertical bar indicates the standard error of four spatial**
**replicates. The blue and red lines illustrate the zero-intercept univariate linear regressions by the original and modified models.**
**Each simulated EFd is calculated from the simulated emissions of two nitrogen addition levels (zero and 450 kg N ha$^{-1}$ yr$^{-1}$).**

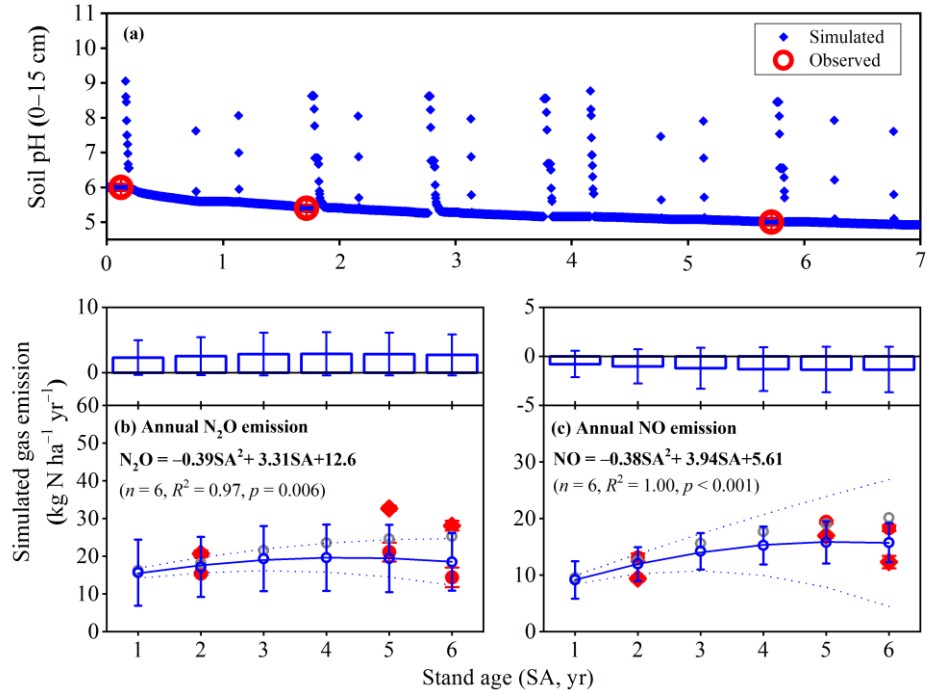


**Figure 5: Simulated topsoil (0−15 cm depth) pH and annual emissions of nitrous oxide (N$_2$O) and nitric oxide (NO) against early**
**tea stand ages by the modified model. The solid lines were the polynomial regression curves. Dashed lines are the lower and upper**
**uncertain bounds at the 95% confidence interval (CI) for regression curves. Each pH datum is given as the daily mean of eight**
**diurnal simulations (3 h for each). The vertical bar crossing each datum point in panel b or c represents the uncertainty (95% CI)**
**induced by those of model inputs. Each box above panels b−c represents total model error that was estimated by referring to mean**
**of model relative errors (MRBs), with vertical bars representing the uncertainties (95% CI) estimated by referring to the double**
**standard deviations of |MRBs|. The red circle and diamond points in panel b or c represent the observed emissions of N$_2$O and NO**
**from urea and organic manure treatments. The grey circle point in panel b or c represents the simulation by the original model.**

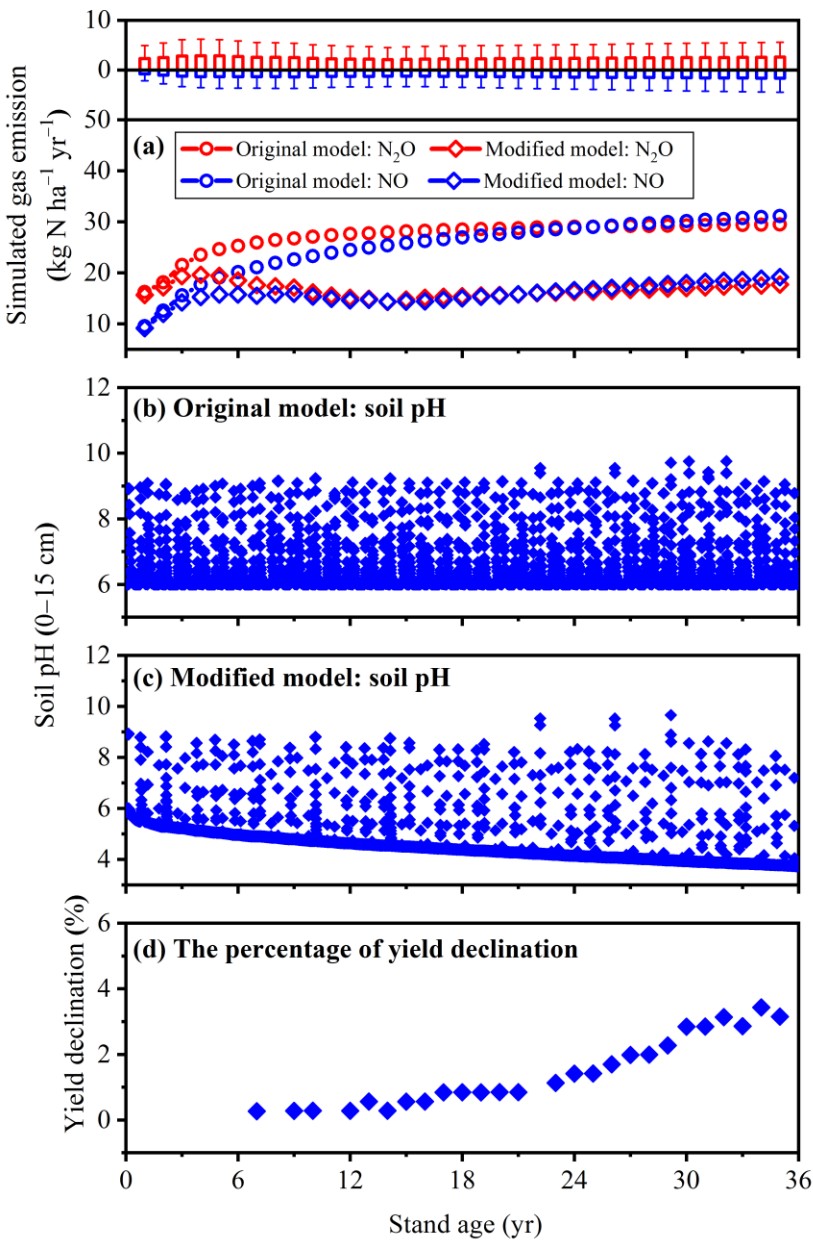


**Figure 6: Simulated emissions of nitrous oxide (N$_2$O) and nitric oxide (NO) and topsoil (0−15 cm) pH of a urea-fertilized tea**
**plantation against stand ages within full lifetime of tea (35 years). Each box above panel a represents total model error of the**
**simulated emissions by the modified model that was estimated by referring to mean of model relative errors (MRBs), with vertical**
**bars representing the uncertainties (95% CI) estimated by referring to the double standard deviations of |MRBs|. The given**
**percentage of yield declination, simulated by the modified model, was due to the effect of soil pH on tea growth.**