# Peer review of "Effects of fertilization and stand age on N2O and NO emissions from tea plantations: A site-scale study in a subtropical region using a modified biogeochemical model"

_Atmospheric Chemistry and Physics, 2019_

## Referee Comment (RC1) · Anonymous Referee #2 · 24 Mar 2020

This manuscript presents an outstanding study. First, the authors modified the process-oriented biogeochemical model: CNMM-DNDC by adding tea growth-related processes that could induce a soil pH reduction and simulate NO and N2O emissions under very low soil pH condition. This is an very important improvement of the CNMM-DNDC model. Second, the authors made in-situ measurements on NO and N2O fluxes in tea planted soil and calibrated the emission factors for both NO and N2O. Therefore the improved CNMM-DNDC model could be used for quantifying NO and N2O emissions under tropical and subtropical low pH soils. The study is perfect to fit the scope

of the ACP journal. I recommend to accept it in its current form.

---

## Referee Comment (RC2) · Anonymous Referee #1 · 31 Mar 2020

General comments: This study modified a process-based model to investigate effects of fertilization and stand age on N2O and NO emissions from tea plantations. The authors did intensive works, including model modification, model evaluation, scenario analysis, and uncertainty analysis. However, the authors provided limited discussions for some important contents/results (please refer specific comments) and some descriptions were not very clear. I suggest author further improving the language, clarity, and discussions of the manuscript.

Specific comments: Lines 21 to 22: This sentence is not clear. For example, did you

mean the observations showed a 62% increase and the simulations showed a 36% increase? Maybe not using "consistently" in this case. Line 31: The "word" should be "world". Line 60: The "which" is not clear; suggest rewriting. Line 113: "full" application is not as clear as "exclusive application". Section 2.2: My understanding is that soil buffering impact is an important mechanism regulating soil H and pH changes. But it looks that there was no parameterization of this mechanism from the equations and parameters described in this section. Line 169: The "tea residue" was actually "tea leaf residue"; right? Lines 217 to 220: This sentence is too long; I suggest breaking this long sentence. Lines 231 to 235: This sentence is long and not clear for me. Did you mean you have set 35 different scenarios for investigating the stand age effects? Line 256: Please add refs for the latin hypercube sampling. In addition, I cannot well capture the author's intention for breaking simulation error into structure error and input error after reading the manuscript. If the break is important, I suggest briefly introducing the intention and discussing the breaking results to better guide readers. Lines 286 to 289: This sentence is not clear. Not sure which values are for the original model and for the modified model. Lines 298 to 302: Given the observed effects (e.g., -25%) and the relatively large observational errors (e.g., 73%), I am curiosity about if the observed stimulation or mitigation effects were significant or not. Lines 305 to 307: It looks (table 6) that the original model performed better than the modified model in capturing the inhibitory effects on NO fluxes; right? This need to be mentioned and discussed. Lines 320 and 322: Seems the original model performed better in simulating the EFds of NO. Could you provide an explanation? Lines 327 to 329: This sentence is not clear; please rewrite. Lines 342 to 343: It is not straightforward to compare the field observations with the uncertainty of the regression lines. Why did not directly compare with the uncertainty of the simulations? Line 353: It looks the differences in simulating gas fluxes and yields between the original and modified models were primarily induced by pH differences. Did you find any observations or literatures that reported soil pH (or soil pH change) in old tea fields (such as 35-years)? This kind of reports could increase the reliability of simulating soil pH change by the modified model. Discussion section: One

important part of this manuscript is model modifications. However, limited discussions regarding the model modifications were provided. I suggest authors providing more discussions regarding the model modifications; such as implications, limitations, and advantages of the modified model etc. Lines 387 to 388: I suggest deleting this sentence because the former results (i.e. comparable models performance for the early stand ages and no-validated models performance for the old stand ages) cannot confirm this conclusion. Lines 397 to 398: Please change the word of "dominant" as it may be not proper to call "62 to 67%" and "57 to 62%" as dominant. Lines 498 to 499: As I previously mentioned, I don't think this conclusion is solid since the long term N2O and NO emissions were not evaluated. Figure 3: The blue cycles are for the modified model and the gray cycles are for the original model?

---

## Author Comment (AC1) · 8 Apr 2020

**Responses to reviews (RC2)**

Anonymous Referee #1

General comments: This study modified a process-based model to investigate effects of fertilization and stand age on N2O and NO emissions from tea plantations. The authors did intensive works, including model modification, model evaluation, scenario analysis, and uncertainty analysis. However, the authors provided limited discussions for some important contents/results (please refer specific comments) and some descriptions were not very clear. I suggest author further improving the language, clarity, and discussions of the manuscript.

**Thanks very much for the evaluation of this manuscript. Based on the reviewer's constructive suggestions, the manuscript has been revised by rewriting the long sentences, updating the inexplicit descriptions and adding new parts of discussion about model modifications and performances for improving the language, clarity and discussions of the manuscript. For the details, please refer to the specific responses.**

Specific comments:

Lines 21 to 22: This sentence is not clear. For example, did you mean the observations showed a 62% increase and the simulations showed a 36% increase? Maybe not using "consistently" in this case.

**Revised**. The description of this sentence has been revised as the reviewer suggested. "*For the modified model, the observations and simulations demonstrated that short-term replacement of urea with oilcake stimulated $N_2O$ emissions by ~62% and ~36% and mitigated NO emissions by ~25% and ~14%, respectively.*" (**Please see lines 21-22**).

Line 31: The "word" should be "world".

**Revised**. The sentence has been revised as the reviewer suggested. "*Tea (Camellia sinensis (L.) Kuntze), as a perennial cash crop, has been widely cultivated long-term in the tropical and subtropical regions of the world*" (**Please see line 31**).

Line 60: The "which" is not clear; suggest rewriting.

**Revised**. The sentence has been revised by clarifying the meaning of "which" as the reviewer suggested. "*In 2013, for instance, the $N_2O$ emissions from tea plantations in China accounted for more than one-tenth of the national total emissions of this gas from croplands and contributed to 85% of the total $N_2O$ emissions from global tea plantations (Li et al., 2016).*" (**Please see lines 59-61**).

Line 113: "full" application is not as clear as "exclusive application".

**Revised**. The sentence has been revised as the reviewer suggested by using "exclusive application" instead of "full application" (**Please see line 113**).

Section 2.2: My understanding is that soil buffering impact is an important mechanism regulating soil H and pH changes. But it looks that there was no parameterization of this mechanism from the equations and parameters described in this section.

Revised. The equations for the parameterization of soil buffering impact has been added in Table S3 as the reviewer suggested "*The soil pH dynamics affected by the urea hydrolysis, soil buffering and manure application have already been considered in the original CNMM-DNDC (Table S3).*" (**Please see lines 182-183** and **Table S3**).

Line 169: The "tea residue" was actually "tea leaf residue"; right?

Revised. The meaning of "tea residue" has been explained as the reviewer suggested which was mainly tea leaf residue and also indicated some young branch due to trimming. "*the decomposition of tea residues due to the trimming (tea leaves and young branch) or falling of old leaves*" (**Please see lines 143-144**).

Lines 217 to 220: This sentence is too long; I suggest breaking this long sentence.

Revised. The sentence has been revised as the reviewer suggested by breaking the long sentence into two sentences. "*The virtual experiments were designed to evaluate the performance of the original and modified models in simulating the annual $EF_d$s and to investigate the effects of fertilizer nitrogen doses on $EF_d$s. For each field treatment exclusively applied with urea or oilcake in 2013 or 2014, virtual experiments against nitrogen addition rates varying from zero to 600 (with an interval of 50) kg N ha$^{-1}$ yr$^{-1}$ were carried out.*" (**Please see lines 220-223**).

Lines 231 to 235: This sentence is long and not clear for me. Did you mean you have set 35 different scenarios for investigating the stand age effects?

Revised. The sentence has been revised as the reviewer suggested by breaking the long sentence into two sentences and clarifying the setting of 35 independent scenarios. "*To ensure the simulations of all the stand ages can be driven by the same meteorological conditions that would be the same as the measured data during the year-round period from September 17$^{th}$, 2013 to October 9$^{th}$, 2014, 35 independent scenarios were designed. Thus, the seedling transplanting for the stand ages of 35, 34, ..., 1 year were set to occur in March of 1979, 1980, ..., and 2013, respectively.*" (**Please see lines 235-238**).

Line 256: Please add refs for the latin hypercube sampling.

Revised. The reference has been added for the Latin hypercube sampling as the reviewer suggested "*It was estimated using the Monte Carlo test with Latin hypercube sampling (Helton and Davis, 2003)*" and "*Helton, J. C., and Davis, F. J.: Latin hypercube sampling and the propagation of uncertainty in analysis of complex systems, Reliab. Eng. Syst. Saf., 81, 23-69, 2003.*" (**Please see lines 260 and 596-597**).

In addition, I cannot well capture the author's intention for breaking simulation error

into structure error and input error after reading the manuscript. If the break is important, I suggest briefly introducing the intention and discussing the breaking results to better guide readers.

**Revised**. The result and discussion about the total model error and input error have been added as the reviewer suggested "*Compared with the uncertainty induced by the inputs ($\varepsilon_{input}$), the absolute values of the total model uncertainty ($\varepsilon_s$) were much smaller, which only accounted for 32% and 35% of the $\varepsilon_{input}$ for $N_2O$ and $NO$, respectively.*" and "*In addition, the smaller total model uncertainty, which only accounts for 33% of the uncertainty induced by inputs, indicated that increasing the reliability of the inputs of soil properties and plant growth parameters can improve the model efficiency.*" (**Please see lines 350-351 and 505-507**).

Lines 286 to 289: This sentence is not clear. Not sure which values are for the original model and for the modified model.

**Revised**. The sentence has been revised as the reviewer suggested. "*For the original model, three (NO) and five ($N_2O$) of the nine individual simulations for each gas showed |MRBs| larger than the corresponding observed two times CV, while |MRBs| larger than the observed two times CV were four (NO and $N_2O$) for the modified model (Table S5). However, the statistics of both models still indicated agreements for annual emissions, with the IA and NSI values of 0.96−0.98 and 0.81−0.94, respectively, for $N_2O$ and NO (Table 1).*" (**Please see lines 290-294**).

Lines 298 to 302: Given the observed effects (e.g., -25%) and the relatively large observational errors (e.g., 73%), I am curiosity about if the observed stimulation or mitigation effects were significant or not.

**Revised**. The observed stimulation and mitigation effects were significant which has been added as the reviewer suggested. In addition, different from the standard error presented in observations, two times standard deviation was applied for calculating the relative observational error of urea replacement effects. "*Based on the statistical analysis using linear mixed models, both the stimulation and mitigation effects were significant (p < 0.05) (Yao et al., 2015).*" and "*$\delta_o$ and $\delta_u$ (in kg N ha$^{-1}$ yr$^{-1}$) signify the corresponding observational errors in two times standard deviation (SD)*" (**Please see lines 305-306 and 217-218**).

Lines 305 to 307: It looks (table 6) that the original model performed better than the modified model in capturing the inhibitory effects on NO fluxes; right? This need to be mentioned and discussed. Lines 320 and 322: Seems the original model performed better in simulating the EFds of NO. Could you provide an explanation?

**Revised**. A new paragraph has been added to discuss the simulation of annual emissions by the original and modified models (**Please see lines 407-420**). The unsatisfactory performance of the modified model for the NO inhibitory effects and EF$_d$s has been discussed as the reviewer suggested. "*The underestimation (2.56 kg N ha$^{-1}$ yr$^{-1}$) and overestimation (3.29 kg N ha$^{-1}$ yr$^{-1}$) of the NO emission in 2014 for T08-UN and T08-OM, respectively, resulted in the significant underestimation of the*

*inhibition effects and increased model relative bias for the modified model. The inhibited NO emissions were also partly attributed to the soil heterotrophic nitrification (Yao et al., 2015), which is the direct oxidation of organic nitrogen to nitrate without passing through mineralization. However, the heterotrophic nitrification was not considered in the model, which may result in the overestimated NO emissions in 2014 for the manure treatments by both models. In addition, compared with the original model, the underestimated NO emission mentioned above was also the key reason for the unsatisfactory simulation of $EF_ds$, which led to the increment of the ZIR slope by 8% (1.0 for the ZIR without T08-UN and 1.08 for the ZIR with T08-UN)."* (**Please see lines 411-418**).

Lines 327 to 329: This sentence is not clear; please rewrite.
**Revised**. The sentence has been rewrite as the reviewer suggested. "*The original model simulations of annual $N_2O$ and NO emissions showed |MRB| of ~33% (ranging from 6−76%) and ~6% (ranging from 3−10%) respectively, while |MRBs| of the annual $N_2O$ and NO emissions were ~17% (ranging from 11−28%) and ~8% (ranging from 1−14%) for the modified model.*" (**Please see lines 333-335**).

Lines 342 to 343: It is not straightforward to compare the field observations with the uncertainty of the regression lines. Why did not directly compare with the uncertainty of the simulations?
**Revised**. The description has been updated as the reviewer suggested. "*As Figure 5 indicated, almost all the field observations in the fertilized fields not only generally fell within the range of the uncertainty induced by the input items, but also within the upper and lower bounds of uncertainty (95% CI) of the regressions.*" (**Please see lines 348-350**).

Line 353: It looks the differences in simulating gas fluxes and yields between the original and modified models were primarily induced by pH differences. Did you find any observations or literatures that reported soil pH (or soil pH change) in old tea fields (such as 35-years)? This kind of reports could increase the reliability of simulating soil pH change by the modified model.
**Revised**. According to the suggestion of the reviewer, the discussion of soil pH changes over 35-years has been added based on the reported average annual soil pH declination rate of tea plantations in Jiangsu and Anhui provinces. "*However, the studies of the tea plantations in Jiangsu and Anhui provinces showed that the average soil pH (0−20 cm) declination rate was 0.06 pH $yr^{-1}$ (Luo, 2006; Su, 2018). For the simulation of 35-year tea plantation in this study, the calculated average annual soil (0−20 cm) pH declination rate was close to the reports with the value of 0.064. Therefore, the consistent declination rate indicates the modifications improve the scientific mechanisms of the biogeochemical model which could be applied for long time simulation.*" (**Please see lines 377-382**).

Discussion section: One important part of this manuscript is model modifications.

However, limited discussions regarding the model modifications were provided. I suggest authors providing more discussions regarding the model modifications; such as implications, limitations, and advantages of the modified model etc.

**Revised**. According to the suggestion of the reviewer, a section of discussion about model modifications has been added. "*The modified CNMM-DNDC was hypothesized to reflect the general knowledge that tea can grow in soils with a suitable pH within 4.0−6.5 (Cao et al., 2009). But the transient increase of soil pH due to urea hydrolysis has no impact on plant growth, as the soil pH could be recovered within a few days due to soil buffering effect. Due to the lack of observed tea yields, the parameterized impact of soil pH on tea growth could not be calibrated or validated in this study, but virtual experiments showed increased yield reduction with increasing stand age, implicating the intensified negative effects on plant growth for older tea plantations. The newly added scientific processes relating to pH reduction were calibrated using the observed soil pH for different stand ages during the early stage of a tea plantation. Although the simulations showed that the modified CNMM-DNDC with the calibrated parameters could accurately reflect the basal soil pH declination during the early years, validation was still missing due to a lack of available independent observation of pH. However, the studies of the tea plantations in Jiangsu and Anhui provinces showed that the average soil pH (0−20 cm) declination rate was 0.06 pH yr$^{-1}$ (Luo, 2006; Su, 2018). For the simulation of 35-year tea plantation in this study, the calculated average annual soil (0−20 cm) pH declination rate was close to the reports with the value of 0.064. Therefore, the consistent declination rate indicates the modifications improve the scientific mechanisms of the biogeochemical model which could be applied for long time simulation. As the actual soil pH would not decline constantly (Yao et al., 2018), the validation of soil pH dynamics for long time is still necessary. The simulated annual emissions by both models were comparable in the early tea stand ages, but those by the modified model were much lower in the mid to late stages of tea lifetime. According to the modifications, the different annual emissions of both gases should be primarily attributed to the soil pH differences. Therefore, the proper simulation of soil pH declination for long time increased the reliability of the simulated variation of annual emissions even though validation of the differences was still missing due to lacking of field observations. Thus, further study is still needed to confirm the general model applicability, especially for the simulations of long term yields, soil pH dynamics, N$_2$O and NO emissions from tea plantations subject to different conditions.*" (**Please see lines 369-389**).

Lines 387 to 388: I suggest deleting this sentence because the former results (i.e. comparable models performance for the early stand ages and no-validated models performance for the old stand ages) cannot confirm this conclusion.

**Revised**. The sentence has been deleted as the reviewer suggested.

Lines 397 to 398: Please change the word of "dominant" as it may be not proper to call "62 to 67%" and "57 to 62%" as dominant.

**Revised**. The word of "dominant" has been revised as "major" as the reviewer

suggested (**Please see line 428**). "

Lines 498 to 499: As I previously mentioned, I don't think this conclusion is solid since the long term N2O and NO emissions were not evaluated.
**Revised**. The sentence of the uncertain conclusion has been deleted and the content of conclusion has been adjusted as the reviewer suggested (**Please see lines 520-547**).

Figure 3: The blue cycles are for the modified model and the gray cycles are for the original model?
**Revised**. The footnotes for the grey and blue circles in panels c−f have been added in Figure 3 as the reviewer suggested (**Please see Figure 3**).

---

## Author Comment (AC2) · 8 Apr 2020

Responses to reviews (RC2) Anonymous Referee #1

General comments: This study modified a process-based model to investigate effects of fertilization and stand age on N2O and NO emissions from tea plantations. The authors did intensive works, including model modification, model evaluation, scenario analysis, and uncertainty analysis. However, the authors provided limited discussions for some important contents/results (please refer specific comments) and some de-

scriptions were not very clear. I suggest author further improving the language, clarity, and discussions of the manuscript.

Thanks very much for the evaluation of this manuscript. Based on the reviewer's constructive suggestions, the manuscript has been revised by rewriting the long sentences, updating the inexplicit descriptions and adding new parts of discussion about model modifications and performances for improving the language, clarity and discussions of the manuscript. For the details, please refer to the specific responses.

Specific comments: Lines 21 to 22: This sentence is not clear. For example, did you mean the observations showed a 62% increase and the simulations showed a 36% increase? Maybe not using "consistently" in this case.

Revised. The description of this sentence has been revised as the reviewer suggested. "For the modified model, the observations and simulations demonstrated that short-term replacement of urea with oilcake stimulated $N_2O$ emissions by $\sim$62% and $\sim$36% and mitigated NO emissions by $\sim$25% and $\sim$14%, respectively".

Line 31: The "word" should be "world".

Revised. The sentence has been revised as the reviewer suggested. "Tea (Camellia sinensis (L.) Kuntze), as a perennial cash crop, has been widely cultivated long-term in the tropical and subtropical regions of the world".

Line 60: The "which" is not clear; suggest rewriting.

Revised. The sentence has been revised by clarifying the meaning of "which" as the reviewer suggested. "In 2013, for instance, the $N_2O$ emissions from tea plantations in China accounted for more than one-tenth of the national total emissions of this gas from croplands and contributed to 85% of the total $N_2O$ emissions from global tea plantations (Li et al., 2016)".

Line 113: "full" application is not as clear as "exclusive application".

Revised. The sentence has been revised as the reviewer suggested by using "exclusive application" instead of "full application".

Section 2.2: My understanding is that soil buffering impact is an important mechanism regulating soil H and pH changes. But it looks that there was no parameterization of this mechanism from the equations and parameters described in this section.

Revised. The equations for the parameterization of soil buffering impact has been added in Table S3 as the reviewer suggested "The soil pH dynamics affected by the urea hydrolysis, soil buffering and manure application have already been considered in the original CNMM-DNDC (Table S3)".

Line 169: The "tea residue" was actually "tea leaf residue"; right?

Revised. The meaning of "tea residue" has been explained as the reviewer suggested which was mainly tea leaf residue and also indicated some young branch due to trimming. "the decomposition of tea residues due to the trimming (tea leaves and young branch) or falling of old leaves".

Lines 217 to 220: This sentence is too long; I suggest breaking this long sentence.

Revised. The sentence has been revised as the reviewer suggested by breaking the long sentence into two sentences. "The virtual experiments were designed to evaluate the performance of the original and modified models in simulating the annual EFds and to investigate the effects of fertilizer nitrogen doses on EFds. For each field treatment exclusively applied with urea or oilcake in 2013 or 2014, virtual experiments against nitrogen addition rates varying from zero to 600 (with an interval of 50) kg N ha$-1$ yr$-1$ were carried out".

Lines 231 to 235: This sentence is long and not clear for me. Did you mean you have set 35 different scenarios for investigating the stand age effects?

Revised. The sentence has been revised as the reviewer suggested by breaking the long sentence into two sentences and clarifying the setting of 35 independent scenarios. "To ensure the simulations of all the stand ages can be driven by the same meteorological conditions that would be the same as the measured data during the year-round period from September 17th, 2013 to October 9th, 2014, 35 independent scenarios were designed. Thus, the seedling transplanting for the stand ages of 35, 34, ..., 1 year were set to occur in March of 1979, 1980, ..., and 2013, respectively".

Line 256: Please add refs for the latin hypercube sampling.

Revised. The reference has been added for the Latin hypercube sampling as the reviewer suggested "It was estimated using the Monte Carlo test with Latin hypercube sampling (Helton and Davis, 2003)" and "Helton, J. C., and Davis, F. J.: Latin hypercube sampling and the propagation of uncertainty in analysis of complex systems, Reliab. Eng. Syst. Saf., 81, 23-69, 2003".

In addition, I cannot well capture the author's intention for breaking simulation error into structure error and input error after reading the manuscript. If the break is important, I suggest briefly introducing the intention and discussing the breaking results to better guide readers.

Revised. The result and discussion about the total model error and input error have been added as the reviewer suggested "Compared with the uncertainty induced by the inputs ($\varepsilon$input), the absolute values of the total model uncertainty ($\varepsilon$s) were much smaller, which only accounted for 32% and 35% of the $\varepsilon$input for N2O and NO, respectively." and "In addition, the smaller total model uncertainty, which only accounts for 33% of the uncertainty induced by inputs, indicated that increasing the reliability of the inputs of soil properties and plant growth parameters can improve the model efficiency".

Lines 286 to 289: This sentence is not clear. Not sure which values are for the original model and for the modified model.

Revised. The sentence has been revised as the reviewer suggested. "For the original model, three (NO) and five (N2O) of the nine individual simulations for each gas showed |MRBs| larger than the corresponding observed two times CV, while |MRBs| larger than the observed two times CV were four (NO and N2O) for the modified model (Table S5). However, the statistics of both models still indicated agreements for annual emissions, with the IA and NSI values of $0.96-0.98$ and $0.81-0.94$, respectively, for N2O and NO (Table 1)".

Lines 298 to 302: Given the observed effects (e.g., -25%) and the relatively large observational errors (e.g., 73%), I am curiosity about if the observed stimulation or mitigation effects were significant or not.

Revised. The observed stimulation and mitigation effects were significant which has been added as the reviewer suggested. In addition, different from the standard error presented in observations, two times standard deviation was applied for calculating the relative observational error of urea replacement effects. "Based on the statistical analysis using linear mixed models, both the stimulation and mitigation effects were significant ($p < 0.05$) (Yao et al., 2015)." and "$\delta$o and $\delta$u (in kg N ha$-1$ yr$-1$) signify the corresponding observational errors in two times standard deviation (SD)".

Lines 305 to 307: It looks (table 6) that the original model performed better than the modified model in capturing the inhibitory effects on NO fluxes; right? This need to be mentioned and discussed. Lines 320 and 322: Seems the original model performed better in simulating the EFds of NO. Could you provide an explanation?

Revised. A new paragraph has been added to discuss the simulation of annual emissions by the original and modified models. The unsatisfactory performance of the modified model for the NO inhibitory effects and EFds has been discussed as the reviewer suggested. "The underestimation (2.56 kg N ha$-1$ yr$-1$) and overestimation (3.29 kg N ha$-1$ yr$-1$) of the NO emission in 2014 for T08-UN and T08-OM, respectively, resulted in the significant underestimation of the inhibition effects and increased model relative bias for the modified model. The inhibited NO emissions were also partly at-

tributed to the soil heterotrophic nitrification (Yao et al., 2015), which is the direct oxidation of organic nitrogen to nitrate without passing through mineralization. However, the heterotrophic nitrification was not considered in the model, which may result in the overestimated NO emissions in 2014 for the manure treatments by both models. In addition, compared with the original model, the underestimated NO emission mentioned above was also the key reason for the unsatisfactory simulation of EFds, which led to the increment of the ZIR slope by 8% (1.0 for the ZIR without T08-UN and 1.08 for the ZIR with T08-UN)".

Lines 327 to 329: This sentence is not clear; please rewrite.

Revised. The sentence has been rewrite as the reviewer suggested. "The original model simulations of annual N2O and NO emissions showed |MRB| of ∼33% (ranging from 6−76%) and ∼6% (ranging from 3−10%) respectively, while |MRBs| of the annual N2O and NO emissions were ∼17% (ranging from 11−28%) and ∼8% (ranging from 1−14%) for the modified model".

Lines 342 to 343: It is not straightforward to compare the field observations with the uncertainty of the regression lines. Why did not directly compare with the uncertainty of the simulations?

Revised. The description has been updated as the reviewer suggested. "As Figure 5 indicated, almost all the field observations in the fertilized fields not only generally fell within the range of the uncertainty induced by the input items, but also within the upper and lower bounds of uncertainty (95% CI) of the regressions".

Line 353: It looks the differences in simulating gas fluxes and yields between the original and modified models were primarily induced by pH differences. Did you find any observations or literatures that reported soil pH (or soil pH change) in old tea fields (such as 35-years)? This kind of reports could increase the reliability of simulating soil pH change by the modified model.

Revised. According to the suggestion of the reviewer, the discussion of soil pH changes over 35-years has been added based on the reported average annual soil pH declination rate of tea plantations in Jiangsu and Anhui provinces. "However, the studies of the tea plantations in Jiangsu and Anhui provinces showed that the average soil pH (0‒20 cm) declination rate was 0.06 pH yr−1 (Luo, 2006; Su, 2018). For the simulation of 35-year tea plantation in this study, the calculated average annual soil (0‒20 cm) pH declination rate was close to the reports with the value of 0.064. Therefore, the consistent declination rate indicates the modifications improve the scientific mechanisms of the biogeochemical model which could be applied for long time simulation".

Discussion section: One important part of this manuscript is model modifications. However, limited discussions regarding the model modifications were provided. I suggest authors providing more discussions regarding the model modifications; such as implications, limitations, and advantages of the modified model etc.

Revised. According to the suggestion of the reviewer, a section of discussion about model modifications has been added. "The modified CNMM-DNDC was hypothesized to reflect the general knowledge that tea can grow in soils with a suitable pH within 4.0−6.5 (Cao et al., 2009). But the transient increase of soil pH due to urea hydrolysis has no impact on plant growth, as the soil pH could be recovered within a few days due to soil buffering effect. Due to the lack of observed tea yields, the parameterized impact of soil pH on tea growth could not be calibrated or validated in this study, but virtual experiments showed increased yield reduction with increasing stand age, implicating the intensified negative effects on plant growth for older tea plantations. The newly added scientific processes relating to pH reduction were calibrated using the observed soil pH for different stand ages during the early stage of a tea plantation. Although the simulations showed that the modified CNMM-DNDC with the calibrated parameters could accurately reflect the basal soil pH declination during the early years, validation was still missing due to a lack of available independent observation of pH. However, the studies of the tea plantations in Jiangsu and Anhui provinces showed that

the average soil pH (0‒20 cm) declination rate was 0.06 pH yr−1 (Luo, 2006; Su, 2018). For the simulation of 35-year tea plantation in this study, the calculated average annual soil (0‒20 cm) pH declination rate was close to the reports with the value of 0.064. Therefore, the consistent declination rate indicates the modifications improve the scientific mechanisms of the biogeochemical model which could be applied for long time simulation. As the actual soil pH would not decline constantly (Yao et al., 2018), the validation of soil pH dynamics for long time is still necessary. The simulated annual emissions by both models were comparable in the early tea stand ages, but those by the modified model were much lower in the mid to late stages of tea lifetime. According to the modifications, the different annual emissions of both gases should be primarily attributed to the soil pH differences. Therefore, the proper simulation of soil pH declination for long time increased the reliability of the simulated variation of annual emissions even though validation of the differences was still missing due to lacking of field observations. Thus, further study is still needed to confirm the general model applicability, especially for the simulations of long term yields, soil pH dynamics, N2O and NO emissions from tea plantations subject to different conditions".

Lines 387 to 388: I suggest deleting this sentence because the former results (i.e. comparable models performance for the early stand ages and no-validated models performance for the old stand ages) cannot confirm this conclusion.

Revised. The sentence has been deleted as the reviewer suggested.

Lines 397 to 398: Please change the word of "dominant" as it may be not proper to call "62 to 67%" and "57 to 62%" as dominant.

Revised. The word of "dominant" has been revised as "major" as the reviewer suggested.

Lines 498 to 499: As I previously mentioned, I don't think this conclusion is solid since the long term N2O and NO emissions were not evaluated.

Revised. The sentence of the uncertain conclusion has been deleted and the content of conclusion has been adjusted as the reviewer suggested.

Figure 3: The blue cycles are for the modified model and the gray cycles are for the original model?

Revised. The footnotes for the grey and blue circles in panels c−f have been added in Figure 3 as the reviewer suggested.
* * *

---

## Author Response (AR2)

**Responses to editor**

Editor comments

Publish as is.
**Thanks very much for the evaluation and processing of this manuscript.**